# Non-necroptotic MLKL function damages mitochondria and promotes hematopoietic stem cell aging

Yuta Yamada [1,2], Jinjing Yang[2], Akiho Saiki-Tsuchiya [2], Yuji Watanabe[3], Shuhei Koide[2], Shin Murai [4], Yuriko Sorimachi[5], Yu Fukuda [1], Kenta Sumiyama [6], Hiroshi Sagara[3], Hiroyasu Nakano [7,8], Keiyo Takubo [5,9], Atsushi Iwama [2,10] & Masayuki Yamashita [1,2] ✉

Hematopoietic stem cells (HSCs) survive many types of cellular stress but often lose their regenerative and lymphopoietic capacities as a result. Such functional decline also occurs with age, and dysfunctional HSCs with impaired mitochondria accumulate during aging. However, the molecular link between HSC stress response and age-related functional decline remains poorly understood. Here we show that multiple stress responses converge on the RIPK3-MLKL axis to induce age-related changes in HSCs. The necroptosis effector MLKL is readily activated by inflammation and replication stress and accumulates in HSC mitochondria. Consequently, activated MLKL does not cause cell death but impairs HSC self-renewal and lymphoid differentiation. Such MLKL-mediated functional decline also occurs in HSCs during organismal aging, with activated MLKL primarily mediating age-related mitochondrial damage and reduced glycolytic flux. Collectively, our results establish the RIPK3-MLKL axis as a key mediator of HSC aging and identify a necroptosis-independent role of MLKL in mitochondrial damage.

Aging deteriorates hematopoiesis and increases one's susceptibility to hematologic disorders. Although lifelong production of the blood and immune cells relies on the self-renewing ability and multipotency of hematopoietic stem cells (HSCs)[1], compelling evidence suggests that HSCs with reduced regenerative and lymphopoietic potential accumulate with age[2]. Such age-related HSC dysfunction is associated with deleterious changes in their organelles, including accumulation of DNA damage, altered epigenetic regulation, reduced autophagic flux, and impaired mitochondria[3–7], as well as environmental changes such as inflammation and alteration of the HSC-supporting niche cells in the bone marrow (BM)[8–10]. We and others recently showed that restoration of the young niche is insufficient to rejuvenate aged HSC function, highlighting a key role for age-associated cell-intrinsic defects in HSC aging[11,12]. Indeed, recent studies suggest that altered mitochondrial function and metabolism cause functional impairment of aged HSCs[7,13]. Moreover, repeated exposure to inflammatory and replication stress causes an irreversible and age-related deficit in HSC function[14,15]. However, a causal

[1]Division of Experimental Hematology, Department of Hematology, St. Jude Children's Research Hospital, Memphis, TN, USA. [2]Division of Stem Cell and Molecular Medicine, Centre for Stem Cell Biology and Regenerative Medicine, The Institute of Medical Science, The University of Tokyo, Tokyo, Japan. [3]Medical Proteomics Laboratory, The Institute of Medical Science, The University of Tokyo, Tokyo, Japan. [4]Department of Biochemistry, Faculty of Medicine, Toho University, Tokyo, Japan. [5]Department of Stem Cell Biology, National Institute of Global Health and Medicine, Japan Institute for Health Security, Tokyo, Japan. [6]Department of Animal Sciences, Graduate School of Bioagricultural Sciences, Nagoya University, Nagoya, Aichi, Japan. [7]Unit of Host Defense, Faculty of Medicine, Toho University, Tokyo, Japan. [8]Research Administration Organization, Toho University, Tokyo, Japan. [9]Department of Cell Fate Biology and Stem Cell Medicine, Tohoku University Graduate School of Medicine, Sendai, Miyagi, Japan. [10]Laboratory of Cellular and Molecular Chemistry, Graduate School of Pharmaceutical Sciences, The University of Tokyo, Tokyo, Japan. ✉e-mail: masayuki.yamashita@stjude.org

relationship between the cell-intrinsic and cell-extrinsic changes remains undefined.

HSCs are molecularly wired to prevent apoptosis unless otherwise required[16–18], and their apoptosis resistance is further enhanced through age-dependent changes in the environment[19]. We and others previously showed that necroptosis, a form of programmed necrosis that is mediated by the receptor-interacting protein kinase 3 (RIPK3)−mixed lineage kinase like (MLKL) axis and typically occurs in apoptosis-resistant cells, can occur in HSCs and limit their regenerative potential after exposure to proinflammatory ligands and myeloablative stress[20–23]. Indeed, HSCs upregulate the necroptosis effector MLKL via posttranscriptional modification upon inflammation[24], and recent studies suggest a critical role for the RIPK3−MLKL axis in mediating HSC dysfunction after certain stresses and during aging[25–28]. Although these studies attributed RIPK3−MLKL-mediated cell death to functional decline in aged HSCs, dysfunctional HSCs should survive and accumulate in the BM during aging[29], leaving the mechanism whereby the RIPK3−MLKL axis mediates age-related HSC changes unclear. Intriguingly, emerging evidence suggests that MLKL activation does not always induce necroptotic cell death[30–32] but rather can impair membrane-bound organelles, including the endosome, autophagosome, and mitochondria, presumably through disruption of membrane integrity[31,33–35].

Here, we sought to clarify the role of the necroptosis pathway in age-related HSC functional decline. We show that the RIPK3−MLKL axis is readily activated in a subset of HSCs after age-related stresses and reduces HSC self-renewal and lymphopoietic potential without affecting HSC survival.

## Results

### MLKL is preferentially activated in HSCs upon inflammation

Previous studies show that MLKL protein is abundantly expressed in murine immature hematopoietic cells[36] (Supplementary Fig. 1a, b) and the expression level is elevated in HSCs upon polyinosinic-polycytidylic acid (pIC)-induced inflammation[24]. To further assess the activation status of the necroptosis pathway in individual HSCs and other hematopoietic progenitors upon inflammation, we used SMART-Tg mice that ubiquitously express a Förster resonance energy transfer (FRET)-based biosensor that monitors MLKL activation and translocation to the cell membrane[37]. Consistent with our previous observation that necroptosis is selectively activated in HSCs but not in granulocyte/monocyte progenitors during pIC-induced inflammation[22], the SMART signal was increased in HSCs and multipotent progenitors (MPPs) but not in myeloid or lymphoid progenitors upon inflammation induced by administration of pIC, lipopolysaccharide (LPS), or tumor necrosis factor α (TNF-α) (Fig. 1a, b and Supplementary Fig. 1c). This was unexpected, as our previous data indicate that pIC injections induce TNF-α- and nuclear factor κB (NF-κB)-dependent pro-survival signals that protect HSCs from necroptotic death[22]. The FRET signal was increased on day 1 independently of biological sex but disappeared on day 7 post-administration of the proinflammatory ligands (Supplementary Fig. 1d, e). Moreover, the acute FRET activation was significantly diminished in a RIPK3-deficient (Ripk3−/−) background (Fig. 1c and Supplementary Fig. 1f), suggesting that the necroptotic RIPK3−MLKL axis is transiently activated in HSCs in response to inflammation. We transplanted HSCs with high and low FRET signals after LPS treatment (FRET/CFP^hi and FRET/CFP^lo) and observed lower engraftment potential and a tendency toward reduced B-lymphopoietic potential of FRET/CFP^hi HSCs compared to FRET/CFP^lo HSCs (Fig. 1d−g and Supplementary Fig. 1g, h). Collectively, these results indicate that the necroptosis pathway is selectively and transiently activated in HSCs upon inflammation, and its activation status is closely associated with their functional decline.

### A non-necroptotic role of MLKL impairs HSC function

The best-known function of MLKL is to induce necroptotic cell death by plasma membrane rupture. To study the effect of inflammation-induced MLKL activation on HSCs, we assessed changes in HSC numbers in pIC-treated MLKL-deficient (Mlkl−/−) mice. Unexpectedly, phosphatidylserine exposure, a hallmark of the loss of plasma membrane integrity observed prior to cell rupture in necroptosis and apoptosis, was unchanged between wild-type (WT) and Mlkl−/− HSCs, indicating that the effect of inflammation-activated MLKL on HSC survival is negligible (Fig. 2a, b). Accordingly, the absolute numbers of HSCs were comparable between pIC-treated WT and Mlkl−/− mice (Fig. 2c and Supplementary Fig. 2a). As expected, pIC-induced inflammation also caused HSCs to exit quiescence[38] and shift from an endothelial protein C receptor-expressing (EPCR+) self-renewing state[39] to a CD41+ myeloid-primed state[40], but MLKL deficiency had no apparent impact on these changes (Supplementary Fig. 2b−d). However, upregulation of another age-related myeloid-biased HSC marker, neogenin-1 (NEO-1)[41] was significantly attenuated by MLKL deficiency (Fig. 2d). To further assess whether activation of the RIPK3−MLKL axis affects age-related HSC subpopulations, we utilized MLKL-SA2 knock-in (Mlkl^SA2) mice, where MLKL cannot be activated by RIPK3-dependent phosphorylation due to the two serine-to-alanine substitutions at codons 345 and 347 of the Mlkl locus[42], and treated them with PBS or pIC. In addition to NEO-1, we also assessed surface expression of P-selectin and GPR183, which were reported to enrich myeloid-biased[43] and less differentiating[44] subpopulations in aged HSCs, respectively. The results revealed that inactivation of the RIPK3−MLKL axis significantly attenuated pIC-induced upregulation of NEO-1, and to a lesser extent GPR183, but not P-selectin, on HSCs (Fig. 2e and Supplementary Fig. 2e, f). As the total number of HSCs did not differ before and after acute pIC injection (Fig. 2c), this indicates that the RIPK3-MLKL axis promotes transition from NEO-1− lineage-balanced to NEO-1+ myeloid-biased states.

In line with the above MLKL-mediated phenotypic changes, HSC transplantation experiments revealed that whereas MLKL deficiency alone conferred a selective advantage to phosphate-buffered saline-treated control HSCs, MLKL deficiency significantly alleviated a pIC-induced reduction in engraftment and B-lymphopoietic potential (Fig. 2f, g and Supplementary Fig. 2g, h). Moreover, in vivo blockade of RIPK3 kinase activity prior to pIC injection via administration of UH15-38[45] revealed a trend toward ameliorating pIC-induced decline in HSC regenerative capacity and B-lymphoid differentiation potential (Supplementary Fig. 3a−d), suggesting that RIPK3 acts upstream of MLKL to mediate HSC functional decline. Transplantation of NEO-1− and NEO-1+ HSCs isolated from pIC-treated WT and Mlkl^SA2 mice further revealed a trend−despite the limited number of mice−toward improved engraftment of Mlkl^SA2 NEO-1− HSCs, but not NEO-1+ HSCs, compared to WT counterparts (Supplementary Fig. 3e-i), indicating selective protection of NEO-1− lineage-balanced HSCs by blockade of the RIPK3−MLKL axis. This MLKL-mediated effect seems largely cell-intrinsic and independent of environmental necroptosis, as MLKL deficiency had a minor effect on pIC-induced BM inflammation (Supplementary Fig. 3j, k), and pIC-induced suppression of ex vivo EPCR+ HSC expansion was largely attributed to MLKL activity in hematopoietic cells (Supplementary Fig. 3l).

To further investigate the cell death-independent role of MLKL in HSC function, we performed HSC transplantation experiments using Mlkl−/− HSCs reconstituted with N-terminal FLAG-tagged MLKL (FLAG-MLKL), which cannot induce cell death but can oligomerize and translocate to the cell membrane[32,46] (Fig. 2h and Supplementary Fig. 3m). Consistent with the cell-intrinsic, necroptosis-independent regulation of HSCs by MLKL, reconstitution of Mlkl−/− HSCs with FLAG-MLKL significantly reduced their engraftment potential (Fig. 2i−l and Supplementary Fig. 3n). These data collectively indicate that inflammation-induced MLKL activation does not cause HSC

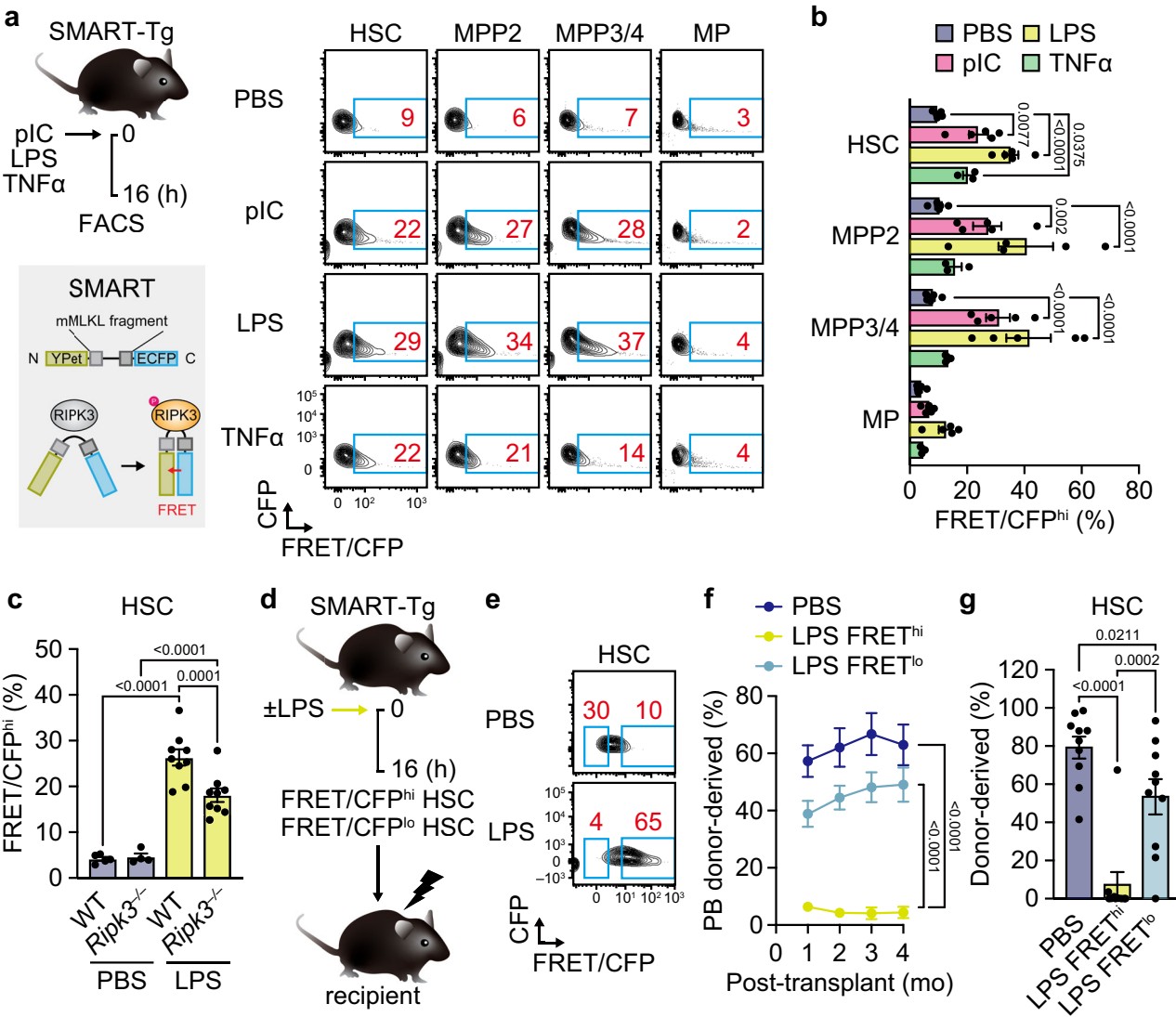

**Fig. 1 | MLKL is preferentially activated in HSCs upon inflammatory stress.**
**a** Experimental design and representative flow cytometry plots for FRET-based detection of MLKL activation in BM HSCs, multipotent progenitors (MPP2 and MPP3/4), and myeloid progenitors (MP). FACS, fluorescence-activated cell sorting. **b** Frequencies of FRET/CFP$^{hi}$ cells in SMART-Tg mice ± pIC, LPS, and TNF-α ($n = 3$ mice in the TNF-α-treated group and 5 mice/other group; two experiments). **c** Frequencies of BM FRET/ CFP$^{hi}$ HSCs in WT and $Ripk3^{-/-}$ SMART-Tg mice ± LPS ($n = 5$ PBS-treated WT, 4 PBS-treated $Ripk3^{-/-}$, 9 LPS-treated WT, and 9 LPS-treated $Ripk3^{-/-}$ mice; two experiments). **d–g** Engraftment potential of BM FRET/CFP$^{hi}$ and

FRET/CFP$^{lo}$ HSCs from LPS-treated SMART-Tg mice. Shown are experimental design (**d**), sorting gates (**e**), PB donor chimerism (**f**) ($n = 10$ PBS-treated, 12 LPS-treated FRET/CFP$^{hi}$, and 12 LPS-treated FRET/CFP$^{lo}$ HSC recipients; two experiments), and donor chimerism in BM HSCs (**g**) ($n = 10$ mice/group; two experiments). HSCs isolated from PBS-treated SMART-Tg mice were used as control donor cells. Data are mean ± s.e.m.; statistical significance was determined using one-way (**c**, **g**) and two-way ANOVA (**b**, **f**) with the two-stage linear step-up procedure of Benjamini, Krieger, and Yekutieli, with exact $P$ values shown.

death but impairs HSC regenerative and lymphopoietic potential independently of MLKL's necroptotic function.

## MLKL mediates age-related HSC changes driven by replication and oncogenic stress

Replication stress is implicated as the driving force of HSC aging[4,15]. Interestingly, MLKL is also preferentially activated in HSCs and MPPs but not in myeloid or lymphoid progenitors upon administration of 5-fluorouracil (5-FU) (Fig. 3a, b and Supplementary Fig. 4a), which induces HSC proliferation and age-related alterations when serially injected[15]. Again, blockade of the RIPK3–MLKL axis did not change

HSC death or their absolute numbers during 5-FU-induced myeloablation (Fig. 3c and Supplementary Fig. 4b–e). Thus, to assess the relevance of MLKL activation in replication-induced HSC aging, we injected WT and $Mlkl^{-/-}$ mice with serial rounds of 5-FU (Fig. 3d). Although hematopoietic recovery after 5-FU injections was comparable between WT and $Mlkl^{-/-}$ mice (Supplementary Fig. 4f), we observed marked attenuation of age-related hematopoietic changes such as myeloid-skewed hematopoiesis (Fig. 3d) and expansion of phenotypic HSCs in 5-FU-treated $Mlkl^{-/-}$ mice (Supplementary Fig. 4g, h). Moreover, HSC transplantation experiments revealed that MLKL deficiency alleviated the 5-FU-induced decrease in HSC

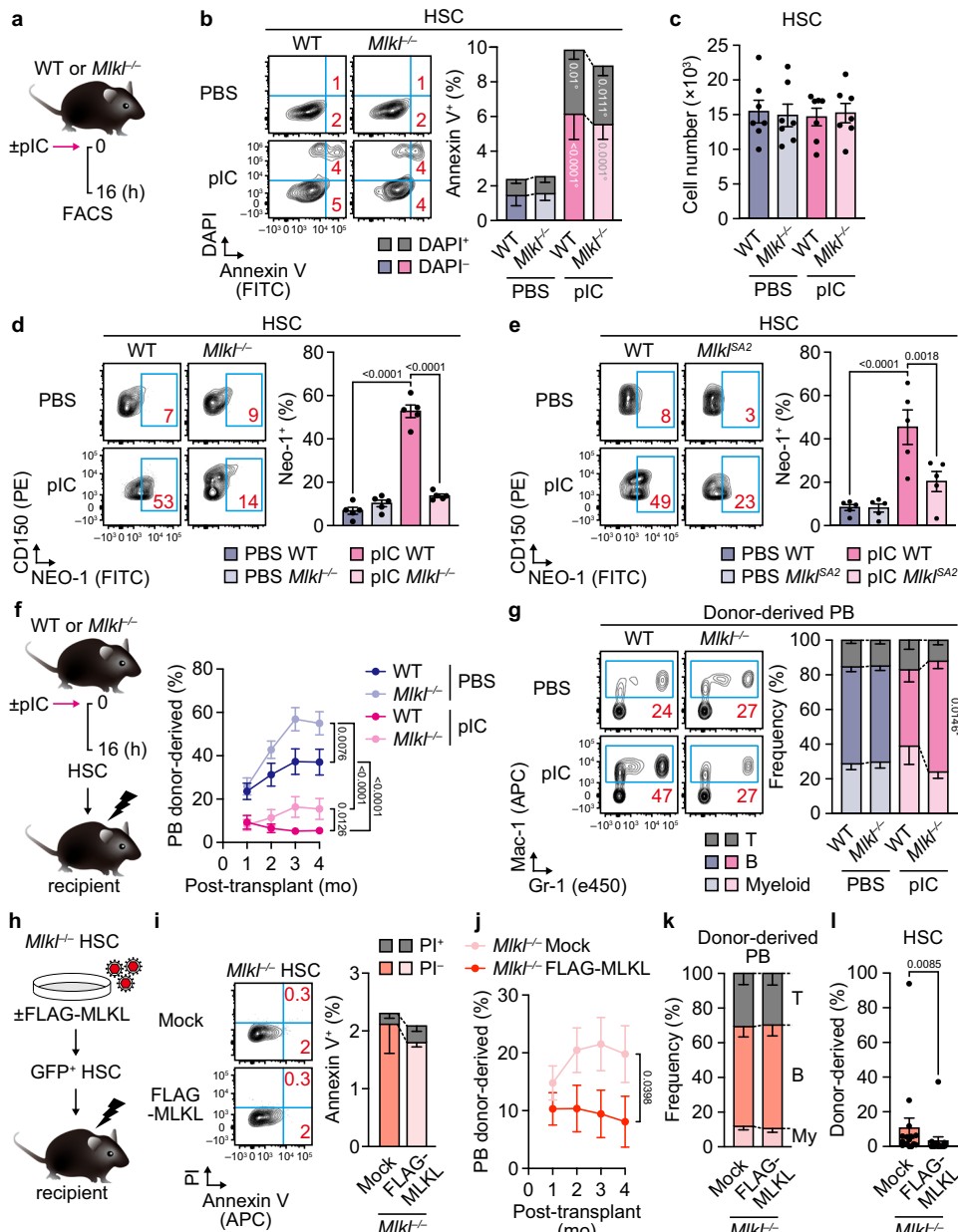

**Fig. 2 | A cell death-independent role of MLKL limits HSC function.**
**a** Experimental design for single pIC injections. **b** Representative flow cytometry plots and frequencies of BM Annexin V⁺ HSCs in WT and *Mlkl⁻/⁻* mice ± pIC (*n* = 7 mice/group; three experiments). **c** Absolute BM HSC numbers in WT and *Mlkl⁻/⁻* mice ± pIC (*n* = 7 mice/group; three experiments). **d** and **e** Representative flow cytometry plots and frequencies of BM NEO-1⁺ HSCs in WT and *Mlkl⁻/⁻* mice (**d**) (*n* = 5 mice/group; one experiment) and WT and *Mlkl^SA2* mice (**e**) (*n* = 5 mice/group; three experiments) ± pIC. **f** and **g** Engraftment potential of BM HSCs from WT and *Mlkl⁻/⁻* mice ± pIC. Shown are experimental design and PB donor chimerism (**f**) (*n* = 14 PBS-treated WT, 12 PBS-treated *Mlkl⁻/⁻*, 14 pIC-treated WT, and 13 pIC-treated *Mlkl⁻/⁻* recipients; three experiments) and donor-derived lineage distribution at 4 months (**g**) (*n* = 14 PBS-treated WT, 12 PBS-treated *Mlkl⁻/⁻*, 8 pIC-treated WT, and 8

pIC-treated *Mlkl⁻/⁻* HSC recipients; three experiments). **h**–**l** Engraftment potential of *Mlkl⁻/⁻* BM HSCs ± N-terminal FLAG-tagged MLKL (FLAG-MLKL). Shown are experimental design (**h**), frequencies of Annexin V⁺ *Mlkl⁻/⁻* BM HSCs ± FLAG-MLKL (**i**) (*n* = 6 pools of 1000 HSCs/group; two experiments), PB donor chimerism (**j**) (*n* = 18 Mock and 17 FLAG-MLKL HSC recipients; three experiments), donor-derived lineage distribution at 4 months (**k**) (*n* = 17 Mock and 12 FLAG-MLKL HSC recipients; three experiments), and donor chimerism in BM HSCs (**l**) (*n* = 16 Mock and 15 FLAG-MLKL HSC recipients; three experiments). My, myeloid. Data are mean ± s.e.m.; statistical significance was determined using two-tailed Mann–Whitney's *U*-test (**l**), one-way ANOVA (**d**, **e**), and two-way ANOVA with the two-stage linear step-up procedure of Benjamini, Krieger, and Yekutieli (**b**, **c**, **f**, **g**, **i**–**k**), with exact *P* values shown; * versus WT; ° versus PBS.

engraftment and lymphopoietic potential (Fig. 3e, f and Supplementary Fig. 4i, j). During serial HSC transplantation, which also induces extensive HSC proliferation in vivo[15], MLKL deficiency in HSCs attenuated a progressive decline in their regenerative potential and lymphoid cell production (Fig. 3g and Supplementary Fig. 5a–c). As an increased propensity to myelodysplastic syndrome (MDS) is another hallmark of aging in the hematopoietic system[47], we next

subjected WT and *Mlkl⁻/⁻* HSCs to oncogenic stress by introducing a C-terminal truncation mutant of Runt-related transcription factor 1 (RUNX1), which mimics MDS in mice by causing myeloid-skewed differentiation and ineffective hematopoiesis[48]. Transplantation of mutant RUNX1–transduced WT or *Mlkl⁻/⁻* HSCs showed that MLKL deficiency in hematopoietic cells did not affect HSC engraftment but attenuated the development of lethal MDS (Fig. 3h and

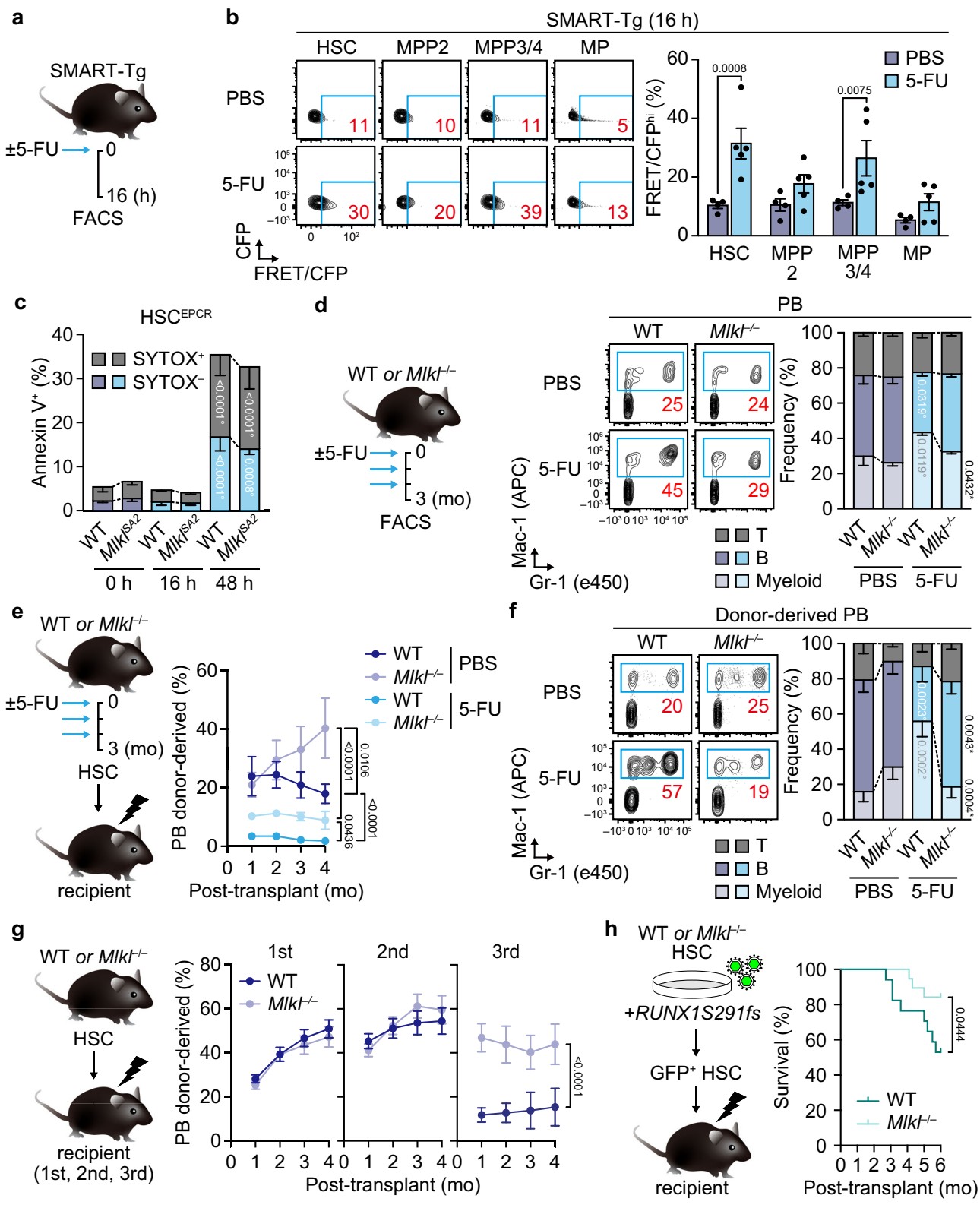

**a** SMART-Tg ±5-FU → 0 ─ 16 (h) FACS

**b** SMART-Tg (16 h)

**c** HSC^EPCR

**d** WT or Mlkl^−/− ±5-FU → 0 ─ 3 (mo) FACS

**e** WT or Mlkl^−/− ±5-FU → 0 ─ 3 (mo) HSC → recipient

**f** Donor-derived PB

**g** WT or Mlkl^−/− HSC → recipient (1st, 2nd, 3rd)

**h** WT or Mlkl^−/− HSC +RUNX1S291fs → GFP⁺ HSC → recipient

Supplementary Fig. 5d). Further analyses of the recipient mice revealed that MLKL deficiency had no detectable impact on mutant RUNX1-induced dysplasia in hematopoietic cells (Supplementary Fig. 5e) but ameliorated lethality mainly due to severe defects in erythropoiesis (Supplementary Fig. 5f). These data support the idea that age-related HSC changes driven by replication and oncogenic stress are mediated at least partly through cell-intrinsic activity of MLKL.

## MLKL mediates HSC dysfunction during organismal aging

The above findings prompted us to investigate whether MLKL mediates the functional decline in HSCs during organismal aging. We

**Fig. 3 | MLKL mediates HSC functional decline upon replication stress.**
**a** Experimental design for single 5-FU (1 × 5-FU) injections. **b** Representative flow cytometry plots and frequencies of FRET/CFP[hi] HSCs, MPPs, and MPs in SMART-Tg mice ± 1 × 5-FU (*n* = 4 PBS-treated and 5 5-FU-treated mice; two experiments). **c** Frequencies of BM Annexin V[+] EPCR[+] HSCs (Lin[−]/CD48[−]/CD150[+]/Sca-1[+]/EPCR[+])[71] in WT and *Mlkl*[SA2] mice at indicated time points following 1 × 5-FU (*n* = 6 mice in the WT 0 h group and 5 mice/other group; two experiments). **d** Experimental design for serial 5-FU injections (3 × 5-FU), representative PB myeloid flow cytometry plots, and lineage distribution in WT and *Mlkl*[−/−] mice ± 3×5-FU (*n* = 4 PBS-treated WT, 4 PBS-treated *Mlkl*[−/−], 3 5-FU-treated WT, and 3 5-FU-treated *Mlkl*[−/−] mice; one experiment). Engraftment potential of BM WT and *Mlkl*[−/−] HSCs ± 3×5-FU (*n* = 4 recipients in the 5-FU-treated *Mlkl*[−/−] group and 5 recipients/other group; one experiment). Shown are experimental design, PB donor chimerism (**e**), and donor-derived lineage distribution at 4 months (**f**). **g** Engraftment potential of WT and *Mlkl*[−/−] BM HSCs after serial transplantation. Shown are experimental design and PB donor chimerism after primary (*n* = 19 WT and 23 *Mlkl*[−/−] recipients; five experiments), secondary (*n* = 25 recipients/group; five experiments), and tertiary transplantation (*n* = 19 WT and 16 *Mlkl*[−/−] recipients; five experiments). **h** *RUNX1S291fs*-transduced MDS mouse models. Shown are experimental design and Kaplan−Meier survival curves for *RUNX1S291fs*-transduced WT and *Mlkl*[−/−] HSC recipients (*n* = 17 WT and 19 *Mlkl*[−/−] recipients; one experiment). Data are mean ± s.e.m.; statistical significance was determined using two-way ANOVA with the two-stage linear step-up procedure of Benjamini, Krieger, and Yekutieli (**b**–**g**) and Mantel−Cox log-rank test (**h**), with exact *P* values shown; * versus WT; ° versus PBS.

analyzed SMART signals in 3- and 12-month-old WT and *Ripk3*[−/−] mice and found that the frequency of FRET/CFP[hi] HSCs increased with age in a RIPK3-dependent manner, particularly in male mice (Fig. 4a−c and Supplementary Fig. 6a). Furthermore, BM analyses of 3- and 18-month-old WT and *Mlkl*[−/−] mice showed that MLKL deficiency largely rescued myeloid-skewed hematopoiesis (Fig. 4d, e) and loss of BM common lymphoid progenitors (Lin[−]/c-Kit[lo]/Sca-1[lo]/Flk2[+]/IL-7Rα[+]) (Supplementary Fig. 6b, c). Although BM HSC numbers and cell death frequency were largely unaffected in aged *Mlkl*[−/−] mice (Supplementary Fig. 6b−d), the accumulation of γH2AX foci, a hallmark of HSC aging, was significantly alleviated in aged *Mlkl*[−/−] HSCs (Fig. 4f and Supplementary Fig. 6e), indicating that the aging process is attenuated in *Mlkl*[−/−] HSCs. Accordingly, transplantation experiments using HSCs of 3- and 18-month-old WT or *Mlkl*[−/−] mice revealed better regenerative and B/T-lymphopoietic potential of aged *Mlkl*[−/−] HSCs compared to aged WT HSCs (Fig. 4g, h and Supplementary Fig. 6f−h), further demonstrating that age-related HSC functional decline is mediated at least partly through an MLKL-dependent mechanism.

MLKL can promote inflammation by releasing damage-associated molecular patterns[21]. Thus, the observed attenuation in HSC aging could be due to the secondary effects of necroptosis-mediated inflammation. To assess this possibility, we examined the levels of various inflammatory cytokines in the BM of 3- and 18-month-old WT and *Mlkl*[−/−] mice. Luminex multiplex cytokine analysis showed no apparent changes in age-associated inflammatory cytokines and chemokines, including interleukin-1 (IL-1) and regulated upon activation normal T cell expressed and secreted (RANTES), which are implicated in myeloid skewing of HSCs[49,50] (Fig. 5a and Supplementary Fig. 7a). In line with this, reciprocal transplantation of 3-month-old WT BM cells into 18-month-old WT and *Mlkl*[−/−] mice revealed no apparent impact of aged *Mlkl*[−/−] environment on myeloid differentiation of WT BM cells (Fig. 5b), suggesting that MLKL minimally affects age-associated BM inflammation.

To further assess the mechanism whereby MLKL mediates HSC aging, we analyzed age-related transcriptomic changes in WT and *Mlkl*[−/−] HSCs (Fig. 5c). Surprisingly, although we could detect differentially expressed genes in WT HSCs with age, including *Selp*, *Mt1*, and *Nupr1* as the top three aging signature genes[43] (Supplementary Fig. 7b, c and Supplementary Data 1−3), few genes were significantly altered between aged WT and *Mlkl*[−/−] HSCs (Fig. 5d−f and Supplementary Data 4). As epigenetic changes can influence HSC fate without transcriptomic changes[51], we also assessed age-related changes in chromatin accessibility of WT and *Mlkl*[−/−] HSCs (Fig. 5g). Again, although many genomic loci showed altered accessibility in WT HSCs with age (Supplementary Fig. 7d, e and Supplementary Data 5−7), few loci were detected as differentially accessible between aged WT and *Mlkl*[−/−] HSCs (Fig. 5h−j and Supplementary Data 8), indicating that MLKL minimally affects the HSC transcriptome and chromatin accessibility during aging.

Together, these results indicate that MLKL mediates HSC dysfunction during organismal aging, likely through mechanisms independent of BM inflammation and transcriptional regulation.

## MLKL mediates age-related mitochondrial damage in HSCs

Despite minimal changes in the transcriptome, gene set enrichment analyses identified several candidate pathways potentially altered in aged *Mlkl*[−/−] HSCs but not in young *Mlkl*[−/−] HSCs (Supplementary Fig. 7f, g). One such candidate was mitochondrial oxidative phosphorylation, which was upregulated in HSCs during aging and closely associated with functional decline in aged HSCs[5]. Indeed, transmission electron microscopy showed that elongated and swollen mitochondria with disorganized cristae, a hallmark of dysfunctional mitochondria observed in HSCs with a history of replication stress[52], were enriched in aged WT but not in aged *Mlkl*[−/−] HSCs (Fig. 6a, b). Consistent with this observation, immune electron microscopy analysis and proximity ligation assays independently revealed an accumulation of active, phosphorylated MLKL at serine 345 (p-MLKL) in the mitochondria of aged HSCs (Fig. 6c, d). Notably, inflammatory exposure and FLAG-MLKL overexpression also induced accumulation of p-MLKL in HSC mitochondria (Supplementary Fig. 8a, b). As previously reported, the level of reactive oxygen species (ROS) in HSCs was reduced with age[5], but no changes in ROS levels were observed between aged WT and *Mlkl*[−/−] HSCs (Supplementary Fig. 8c). Autophagosome frequency and starvation-induced autophagy flux were also similar between aged WT and *Mlkl*[−/−] HSCs (Supplementary Fig. 8d and e), suggesting that MLKL minimally affects autophagy in aged HSCs. However, the age-related decrease in mitochondrial membrane potential was significantly alleviated in aged *Mlkl*[−/−] HSCs (Fig. 6e). Seahorse analyses showed that these changes were associated with improved mitochondrial ATP production and glycolytic flux in aged *Mlkl*[−/−] HSCs compared to aged WT HSCs (Fig. 6f, g and Supplementary Fig. 8f). Although the rarity of HSCs in young animals precluded us from directly comparing mitochondrial metabolism between young and aged *Mlkl*[−/−] HSCs, this could reflect the attenuation of age-related metabolic rewiring from glycolysis to oxidative phosphorylation[5]. In vitro experiments using isolated mitochondria and recombinant MLKL protein revealed that mitochondrial membrane potential was decreased by N-terminal four-helix bundle domain of recombinant mouse MLKL protein (MLKL-NTD) that oligomerizes and forms pores on the cell membrane without stimulation, and to a lesser extent by N-terminal FLAG-tagged MLKL-NTD (FLAG-MLKL-NTD) that retains oligomerization capacity but cannot perform pore formation on the cell membrane, but not by C-terminal pseudokinase domain of recombinant mouse MLKL protein (MLKL-CTD) that does not oligomerize or form pores on the cell membrane[53] (Fig. 6h and Supplementary Fig. 8g), indicating that activated MLKL directly impairs mitochondrial membrane potential via mechanisms both dependent on and independent of its pore-forming capacity. Together, these results identify MLKL as a key mediator of age-related mitochondrial dysfunction and metabolic rewiring in HSCs.

## Discussion

Diverse cellular stresses and naturally occurring organismal aging cause common HSC changes, such as reduced regenerative capacity,

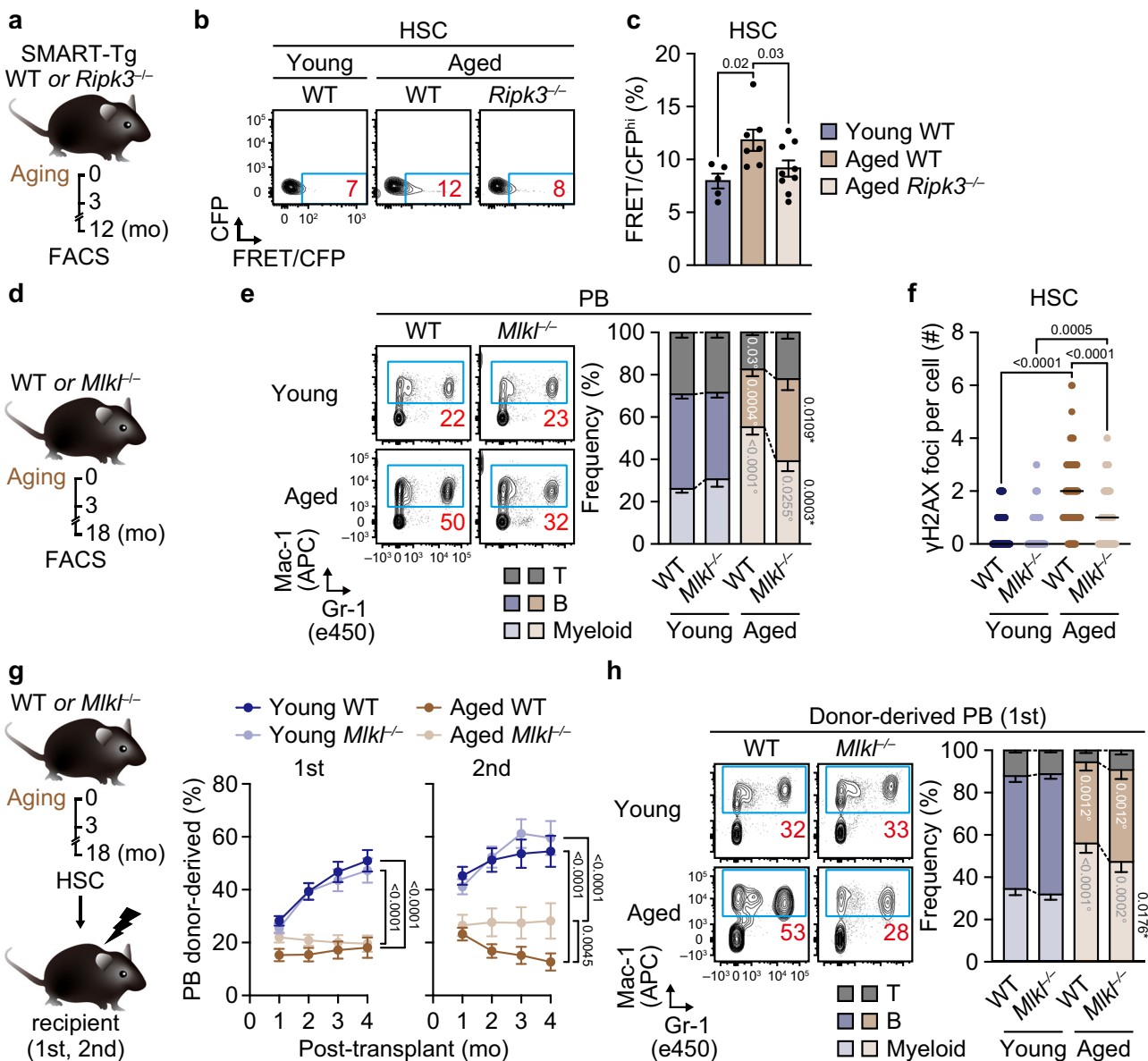

**Fig. 4 | MLKL mediates HSC functional decline during aging. a–c** Activation of the RIPK3-MLKL axis in aged HSCs. Shown are experimental design (**a**), representative flow cytometry plots (**b**), and frequencies of BM FRET/CFPhi HSCs in young WT, aged WT, and aged *Ripk3−/−* SMART-Tg mice (**c**) (n = 5 young WT, 7 aged WT, and 9 aged *Ripk3−/−* SMART-Tg mice; three experiments). Effects of MLKL on hematopoietic aging. Shown are experimental design (**d**), PB lineage distribution (**e**) (n = 9 mice in the aged WT group and 8 mice/other group; five experiments), and the number of γH2AX foci per WT and *Mlkl−/−* HSC ± aging (**f**) (n = 100 cells/group; two experiments). **g** Engraftment potential of BM HSCs from WT and *Mlkl−/−* mice ± aging. Shown are experimental design and donor PB chimerism after

primary (n = 19 young WT, 23 young *Mlkl−/−*, 23 aged WT, and 24 aged *Mlkl−/−* recipients; five experiments) and secondary transplantation (n = 25 young WT, 25 young *Mlkl−/−*, 19 aged WT, and 19 aged *Mlkl−/−* recipients; five experiments). **h** Donor-derived PB lineage distribution in primary recipients of WT and *Mlkl−/−* BM HSCs ± aging at 4 months (n = 19 young WT, 23 young *Mlkl−/−*, 22 aged WT, and 24 aged *Mlkl−/−* recipients; five experiments). Data are mean ± s.e.m. except (**f**), where horizontal lines indicate median; statistical significance was determined using one-way ANOVA (**c**), two-way ANOVA (**e, g, h**), and Kruskal-Wallis test (**f**) with the two-stage linear step-up procedure of Benjamini, Krieger, and Yekutieli, with exact *P* values shown; * versus WT; ° versus young.

myeloid-biased differentiation, and mitochondrial dysfunction, but the shared mechanism is unclear. Here, we identify non-canonical activation of the necroptosis effector MLKL as a mechanism mediating such age-related HSC changes (Supplementary Fig. 8h). Our results show that the response to various age-related stresses, such as inflammation, replication stress, and oncogenic stress, ultimately converges on non-necroptotic activation of the RIPK3−MLKL axis in HSCs. Such stress-induced MLKL activation has little impact on age-related cell-intrinsic changes in HSC survival, the HSC transcriptome, and chromatin accessibility, or cell-extrinsic changes such as BM inflammation. Instead, we show that activated MLKL localizes in mitochondria and directly impairs mitochondrial function, causing an age-related functional decline in HSCs.

Our findings describe a common molecular pathway preferentially activated in HSCs and MPPs to cause cumulative mitochondrial damage. Multiple lines of evidence indicate that HSCs can acquire irreversible cellular changes in response to stress[54]. Lifelong tracking of label-retaining cells suggests that HSCs progressively lose their regenerative

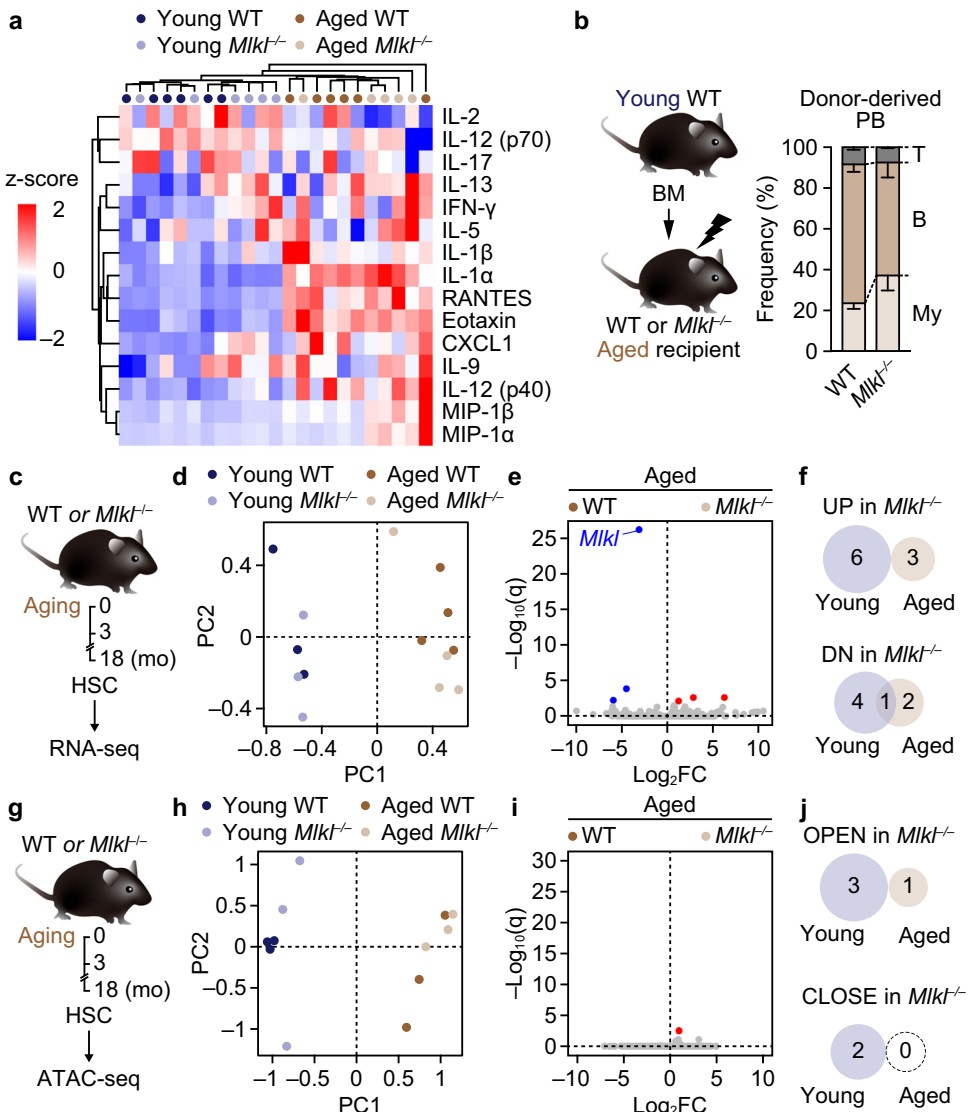

**Fig. 5 | MLKL minimally affects BM inflammation, the HSC transcriptome, and HSC chromatin accessibility during aging. a** Heatmap showing normalized concentration of inflammatory cytokines in the BM of WT and *Mlkl*[−/−] mice ± aging (*n* = 5 mice in the aged *Mlkl*[−/−] group and 6 mice/other group; three experiments). **b** Experimental design and donor-derived PB lineage distribution in aged WT and *Mlkl*[−/−] recipients of young WT BM cells at 4 months post-transplantation (*n* = 6 WT and 8 *Mlkl*[−/−] recipients; one experiment). RNA-seq analysis of WT and *Mlkl*[−/−] BM HSCs ± aging. Shown are experimental design (**c**), a principal component analysis (PCA) plot (**d**), a volcano plot showing differentially expressed genes (DEGs) in aged *Mlkl*[−/−] versus aged WT HSCs (**e**) (*q* < 0.01), and Venn diagrams showing the number of DEGs (**f**) (*n* = 3 young WT, 3 young *Mlkl*[−/−], 4 aged WT, and 4 aged *Mlkl*[−/−] biological replicates; three experiments). Assay for transposase-accessible chromatin (ATAC)-seq analysis of WT and *Mlkl*[−/−] BM HSCs ± aging. Shown are experimental design (**g**), a PCA plot (**h**), a volcano plot showing differentially accessible regions (DARs) in aged *Mlkl*[−/−] versus aged WT HSCs (**i**) (*q* < 0.01), and Venn diagrams showing the number of DARs (**j**) (*n* = 3 biological replicates/group; one experiment). Data are mean ± s.e.m. in (**b**); statistical significance was determined using two-way ANOVA with the two-stage linear step-up procedure of Benjamini, Krieger, and Yekutieli.

potential with age through mechanisms associated with divisional history[55]. Accordingly, repeated exposure to pIC-induced inflammation induces a cumulative inhibitory effect on HSCs with divisional history[14]. Likewise, serial rounds of chemotherapy and transplantation induce an exit from quiescence and cause irreversible age-related HSC dysfunction[15]. Notably, dysfunctional mitochondria accumulate in HSCs after such replication stress and reentry into quiescence, acting as irreversible cell-intrinsic changes that drive HSC dysfunction[52]. Our results align with these observations and further establish that MLKL is commonly activated downstream of such stress responses and mediates mitochondrial damage. Known molecules upstream of RIPK3−MLKL function in HSCs, including RIPK1, Toll-interleukin-1 receptor domain-containing adapter protein inducing interferon beta (TRIF), and Z-DNA binding protein 1 (ZBP1)[20,21,56,57]. These likely account

for the convergent activation of MLKL in HSCs upon exposure to various inflammatory ligands. Our data showed that only a subset of HSCs activate MLKL upon systemic inflammation, which could be due to cell-to-cell differences in the available levels or states of relevant proteins in the heterogenous HSC pool[58] or differences in the microenvironment surrounding HSCs[59]. This, together with the transient nature of MLKL activation after inflammation, supports the hypothesis that intermittently activated MLKL has a cumulative negative impact on a subset of HSCs during organismal aging, eventually causing the accumulation of HSCs with various degrees of dysfunction. Transcription from transposable elements is activated upon replication stress[60,61], which ZBP1 may sense to activate the RIPK3−MLKL axis. DNA damage and imbalanced proteostasis also activate the necroptosis pathway and compromise HSC function[27,28], indicating that many known age-related

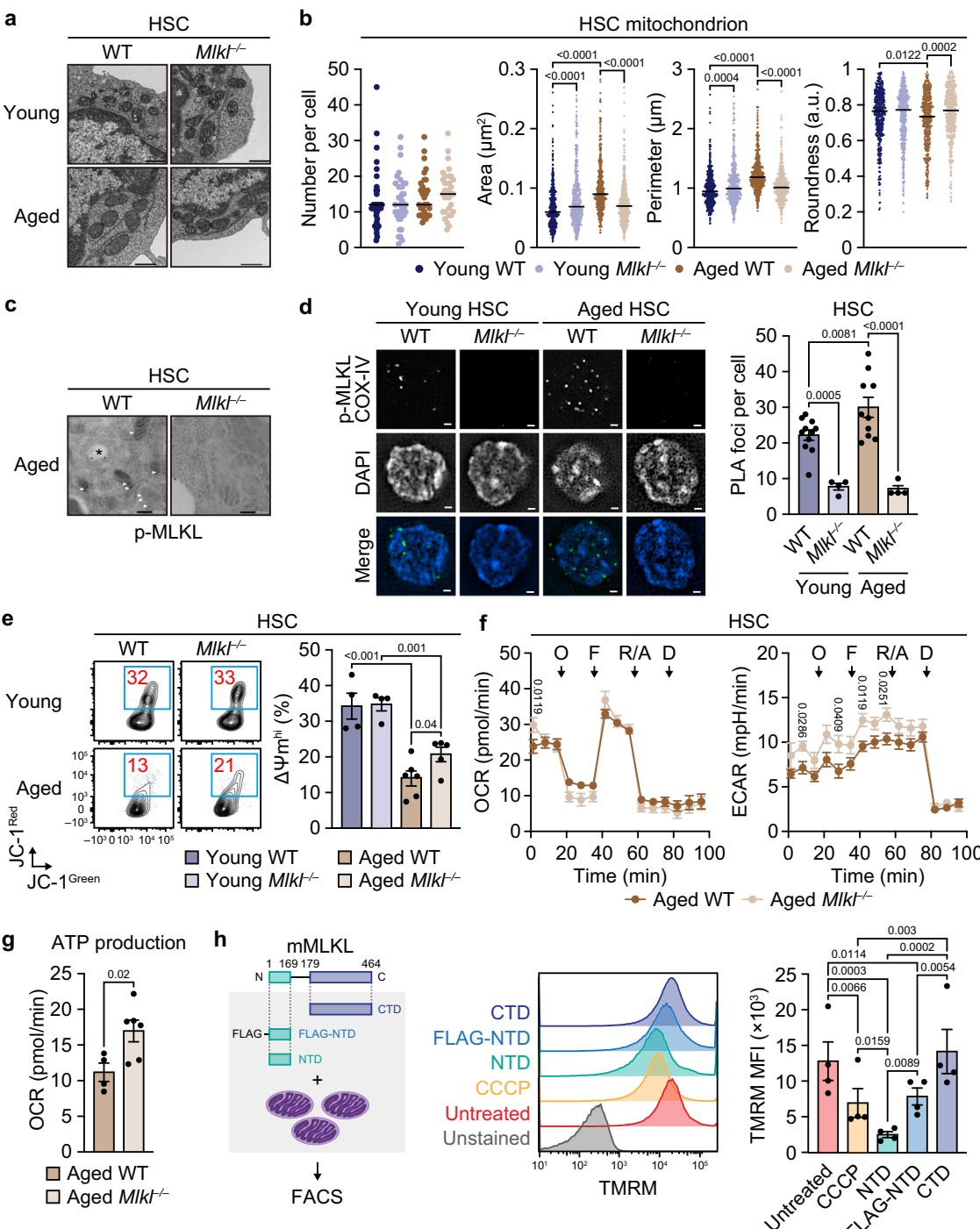

**Fig. 6 | MLKL mediates age-related mitochondrial changes in HSCs.** Transmission electron microscopy (TEM) of BM WT and *Mlkl*−/− HSCs ± aging. Shown are representative images (**a**) and quantification of mitochondrial number/cell, area, perimeter, and roundness (**b**) (*n* = 457 young WT, 484 young *Mlkl*−/−, 450 aged WT, and 513 aged *Mlkl*−/− mitochondria; two experiments). **c** Immunogold TEM for p-MLKL in aged WT and *Mlkl*−/− HSCs. Images are representative of two experiments. Arrowheads and an asterisk (*) indicate signals in mitochondria and endosomes or non-specific signals, respectively. **d** Co-localization of p-MLKL (S345) and COX-IV in HSCs detected by proximity ligation assay (PLA). Shown are representative images and PLA foci per WT and *Mlkl*−/− HSC ± aging (*n* = 11 young WT, 4 young *Mlkl*−/−, 10 aged WT, and 4 aged *Mlkl*−/− HSCs; two experiments). **e** Mitochondrial membrane potential (ΔΨm) in BM WT and *Mlkl*−/− HSCs ± aging. Shown are representative flow cytometry plots and frequencies of BM ΔΨm$^{hi}$ (JC-

1$^{Red+}$) HSCs (*n* = 4 young WT, 4 young *Mlkl*−/−, 6 aged WT, and 5 aged *Mlkl*−/− mice; three experiments). Extracellular flux analysis of aged BM WT and *Mlkl*−/− HSCs (n = 4 WT and 6 *Mlkl*−/− mice; one experiment). Shown are oxygen consumption rate (OCR), extracellular acidification rate (ECAR) (**f**), and ATP-linked respiration (**g**). Arrows indicate injections of oligomycin (O), FCCP (F), rotenone/antimycin (R/A), and 2-deoxy-D-glucose (D). **h** Experimental design, representative flow cytometry plots, and ΔΨm quantification in isolated mitochondria ± recombinant MLKL variants or CCCP (*n* = 4 biological replicates/group; four experiments). Data are mean ± s.e.m. except in (**b**), where horizontal lines indicate median; Scale bars, 500 nm (**a**), 200 nm (**c**), and 1 μm (**d**); Statistical significance was determined using unpaired two-tailed Student's *t*-test (**g**), one-way ANOVA with the two-stage linear step-up procedure of Benjamini, Krieger, and Yekutieli (**b, d, e, h**), and two-way ANOVA with Fisher's LSD test (**f**), with exact *P* values shown.

stressors could ultimately activate MLKL in HSCs. Since emerging evidence suggests other regulators upstream of MLKL than RIPK3 for non-lethal functions[62], whether RIPK3 is absolutely required to cause MLKL-dependent mitochondrial damage in HSCs during organismal aging warrants further study.

Our results highlight a non-necroptotic role of MLKL in age-related HSC dysfunction. Previously, we showed that activation of the TNF-α–p65 axis protects HSCs from necroptotic cell death upon exposure to pIC-induced inflammation[22]. Consistent with this, we did not detect an MLKL-mediated increase in dying and dead HSCs upon pIC treatment, but we did detect activation of MLKL in the surviving HSCs. This reflects a recent study reporting that strong NF-κB activity is a prerequisite for non-lethal activation of the necroptotic pathway in TRAF2-deficient hepatocytes[63]. A pro-survival mechanism that counteracts MLKL-dependent cell death while enabling the localization of activated MLKL in mitochondria, such as ESCRT-III-dependent plasma membrane repair[30,31], should exist in HSCs. Our data indicate that activated MLKL could disrupt mitochondrial function directly by forming pores on the mitochondrial membrane and indirectly by recruiting its binding partners. The latter mechanism could be involved in the observed MLKL-dependent metabolic rewiring from glycolysis to oxidative phosphorylation in HSCs, given that active MLKL recruits RIPK3 to mitochondria and promotes the conversion of pyruvate to acetyl-CoA through the RIPK3-mediated phosphorylation of pyruvate dehydrogenase complex[34]. Our results also suggest that MLKL-dependent HSC dysfunction is likely independent of age-related changes in the transcriptome and chromatin accessibility. Although our bulk sequencing approaches might omit differences that would otherwise be detected with single-cell sequencing, this at least partially explains our previous finding on the limited impact of transcriptomic rejuvenation on functional restoration of already aged HSCs[12] and underscores the importance of investigating molecular mechanisms not involving transcriptional regulation. The partial rescue by MLKL deficiency strongly indicates the existence of other mechanisms that cause mitochondrial damage in HSCs during aging.

Finally, although our results show the detrimental effects of activated MLKL in the context of HSC aging, such a mechanism might be evolutionarily preserved in HSCs because of its beneficial features in other contexts. Although we did not formally analyze the lifespan of *Mlkl*[−/−] mice, we did not recognize any apparent difference in terms of their morbidity and mortality, in line with the prior observation[25]. Given the preferential activation of MLKL in HSCs compared to myeloid progenitors, non-lethal activation of MLKL would enable otherwise dying HSCs to produce sufficient numbers of blood and immune cells to help clear pathogens and tissue debris and resolve the associated inflammation. Indeed, a recent study revealed that loss-of-function mutations in RIPK3 and MLKL are naturally selected for in naked mole rats, which are known for their resistance to age-related tissue dysfunction and cancer development but extreme sensitivity to viral infection[64,65]. Moreover, although we showed that MLKL can accelerate MDS development, others reported that MLKL may instead inhibit leukemia development by promoting the differentiation of leukemia stem cells[66,67]. As factors required for MLKL-dependent necroptosis execution are not necessarily shared between humans and mice, whether our findings in mouse HSCs can directly be applied to human HSCs should be investigated in the future. Understanding these potentially opposing effects by non-lethal activation of MLKL and its relevance to human HSC biology will provide a new opportunity to better control aging of the hematopoietic system and age-related hematologic disorders.

## Methods

### Ethics
All animal experiments comply with all relevant ethical regulations and were performed in accordance with protocols approved by the Animal Care and Use Committee at the Institute of Medical Science, University

of Tokyo (protocol number: PA19-07) and the St. Jude Children's Research Hospital Institutional Animal Care and Use Committee (protocol number: 3252).

### Reagents and resources
Details of key reagents and resources used in this study are listed in Supplementary Table 1.

### Mice
WT CD45.2 and CD45.1 C57BL/6 mice were purchased from Japan SLC and Sankyo Labo Services, respectively. *Mlkl*[−/−] mice were obtained from Dr. M. Pasparakis (University of Cologne)[68], and WT and *Ripk3*[−/−] SMART-Tg mice were obtained from Dr. H. Nakano (Toho University)[37]. *Mlkl*[SA2] mice were purchased from the Jackson Laboratory[42]. WT and *Mlkl*[−/−] mice were bred and aged in-house at the Institute of Medical Science, University of Tokyo, WT and *Ripk3*[−/−] SMART-Tg mice at Toho University, and WT, SMART-Tg, *Mlkl*[−/−], and *Mlkl*[SA2] mice at St. Jude Children's Research Hospital. At the time of analysis, young mice were 6–12 weeks of age, and aged mice were 18–20 months of age unless otherwise specified. For transplantation experiments, 8–12-week–old CD45.1 C57BL/6 mice were used as recipients. Respective littermates or age-matched mice were used as controls. No specific randomization or blinding protocol was used, and both male and female animals were used indiscriminately in this study. All mice were bred and maintained on a 12 h light cycle in temperature- and humidity-controlled specific pathogen-free mouse facilities.

### In vivo assays
For pIC-induced inflammation, mice were injected intraperitoneally with 5 mg/kg pIC (Cytiva) in phosphate-buffered saline (PBS) once or 7 times every other day. For LPS-induced inflammation, mice were injected intraperitoneally with 0.4 mg/kg LPS (InvivoGen) in PBS once. For TNF-α-induced inflammation, mice were injected retro-orbitally with 2 μg TNF-α (Genentech) in PBS once. For repeated 5-FU treatment, mice were injected intraperitoneally with 150 mg/kg 5-FU (Kyowa Kirin) in PBS once a month for 3 months (3 times). For in vivo blockade of RIPK3 kinase activity, mice were injected intraperitoneally with 30 mg/kg UH15-38 (Bio-Techne) in PBS once every day for 4 consecutive days before pIC injection. For HSC transplantation, recipient mice (CD45.1) were lethally irradiated (9.5 Gy, delivered in split doses 3 h apart) using an X-ray irradiator (MBR-1520R; Hitachi) and retro-orbitally injected with 250 donor HSCs (CD45.2[+]) along with $2 \times 10^5$ BM cells (CD45.1[+]) within the next 6 h. When necessary, $2 \times 10^6$ BM cells were pooled from each recipient at 4 months and retro-orbitally injected into freshly prepared, lethally irradiated secondary and tertiary recipients. For BM chimeras, recipient mice were lethally irradiated and retro-orbitally injected with $2 \times 10^6$ BM cells isolated from WT and *Mlkl*[−/−] mice. For reciprocal transplantation, 18-month-old WT and *Mlkl*[−/−] recipient mice were lethally irradiated and retro-orbitally injected with $2 \times 10^6$ BM cells isolated from 3-month-old WT mice. For transplantation of WT and *Mlkl*[−/−] HSCs after RUNX1S291fs transduction, lethally irradiated recipients were retro-orbitally injected with 1000 GFP[+] transduced HSCs along with $5 \times 10^5$ Sca-1-depleted helper BM cells (CD45.1[+]). For transplantation of *Mlkl*[−/−] HSCs reconstituted with N-terminal FLAG-MLKL, lethally irradiated recipients were retro-orbitally injected with 1000 GFP[+] transduced HSCs along with $2 \times 10^5$ BM cells (CD45.1[+]). Transplanted mice received antibiotic water containing Baytril (Bayer) for 4 weeks. Peripheral blood (PB) was analyzed monthly and collected via retro-orbital bleeding in 4 mL of ACK lysis buffer (150 mM NH₄Cl and 10 mM KHCO₃) containing 10 mM EDTA for flow cytometry analyses, and BM was analyzed at 4 months post-transplantation.

### Flow cytometry
Single-cell suspensions of BM cells were obtained in PBS containing 2% heat-inactivated fetal bovine serum (FBS) (Sigma-Aldrich) either by

crushing femurs, tibiae, pelvises, humeri, and the sternum or by flushing femurs and tibiae. Erythrocytes were removed by ACK lysis, and contaminating bone fragments were further removed by centrifugation on a Ficoll gradient (Histopaque-1119; Sigma-Aldrich). BM cellularity was determined by using a ViCELL-XR or ViCELL-BLU automated cell counter (Beckman-Coulter). For immature cell sorting, BM cells were pre-enriched for c-Kit$^+$ cells with c-Kit microbeads and LS columns (Miltenyi Biotec). For immature cell analyses and sorting, a lineage (Lin) cocktail of CD3ε-PECy5 (1:200), CD4-PECy5 (1:800), CD5-PECy5 (1:800), CD8α-PECy5 (1:800), B220-PECy5 (1:800), Mac-1-PECy5 (1:800), Gr-1-PECy5 (1:800), and Ter119-PECy5 (1:400) was used to define immature BM cells. For immature cell sorting, c-Kit-enriched BM cells were stained with Lin-PECy5, c-Kit-APC (1:800), Sca-1-PECy7 (1:800), Flk2-BV421 (1:50), CD48-APCeF780 (1:400), and CD150-PE (1:400). For immature cell analyses, unfractionated BM cells were stained with Lin-PECy5, c-Kit-APC, Sca-1-PECy7, Flk2-Bio (1:100)/SA-BV605 (1:400), CD48-AF700 (1:400), CD150-BV650 (1:200), CD34-FITC (1:50), FcγR-BV510 (1:800), IL-7Rα-APCCy7 (1:100), CD41-BV421 (1:400), and EPCR-PE (1:800). For NEO-1 expression, cells were incubated with anti-FcγR (1 µg per 10$^6$ cells) for 10 min and stained with Lin-PECy5, c-Kit-APC, Sca-1-PECy7, CD48-APCeF780, CD150-PE and anti-NEO-1 (1:13)/anti-goat IgG-AF488 (1:200). For P-selectin and GPR183 expression, cells were stained with Lin-PECy5, c-Kit-APC, Sca-1-PECy7, CD48-APCeF780 together with either CD150-BV650 and P-selectin-PE (1:200) or CD150-PE and GPR183-FITC (1:50). For PB analyses, cells were stained with Ter119-PECy5, Gr-1-eF450 (1:800), Mac-1-APC (1:800), B220-BV605 or -AF700 (1:800), and CD3ε-PECy7 (1:200). For analyses of FRET in immature SMART-Tg cells, unfractionated BM cells were stained with Lin-PECy5, c-Kit-APC, Sca-1-PECy7, CD48-APCeF780, and CD150-PE. For transplantation experiments, PB cells were stained with Ter119-PECy5, Gr-1-eF450, Mac-1-APC, B220-BV605, CD3ε-PECy7, CD45.1-APCeF780 (1:200), and CD45.2-FITC or -BV786 (1:200), and BM cells were stained with Lin-PECy5, c-Kit-APC, Sca-1-PECy7, Flk2-Bio/SA-BV605, CD48-AF700, CD150-BV650, IL-7Rα-PE, CD45.1-APCeF780, and CD45.2-FITC or -BV786. For ex vivo expanded HSC analyses, cells were stained with CD41-FITC (1:800), Mac-1-Bio (1:800)/SA-BV605, FcγR-BV510, Sca-1-PECy7, c-Kit-APC, CD150-BV650, CD48-AF700, and EPCR-PE. For staining with CellROX, unfractionated BM cells were stained with Lin-PECy5, c-Kit-APC, Sca-1-PECy7, Flk2-BV421, CD48-AF700, and CD150-BV650 and incubated with 5 µM CellROX Green in 2% FBS/PBS for 30 min at 37 °C, 5% CO$_2$. For staining with JC-1, unfractionated BM cells were stained with Lin-PECy5, c-Kit-APC, Sca-1-PECy7, Flk2-BV421, CD48-APCeF780, CD150-BV650 and incubated for 30 min at 37 °C, 5% CO$_2$ with 2 µM JC-1 in Iscove's modified Dulbecco's media (IMDM) supplemented with 5% FBS, 1× penicillin–streptomycin–ʟ-glutamine (Fujifilm Wako), 0.1 mM non-essential amino acids (Gibco), 1 mM sodium pyruvate (Gibco), and 50 µM 2-mercaptoethanol (Sigma-Aldrich). All antibody staining was performed on ice for 45 min when a CD34 antibody was included or 30 min otherwise, and stained cells were finally resuspended in 2% FBS/PBS containing 1 µg/ml propidium iodide (PI) to exclude dead cells. For Annexin V staining, unfractionated BM cells stained with Lin-PECy5, c-Kit-APC, Sca-1-PECy7, CD48-APCeF780, CD150-PE, or retrovirally transduced GFP$^+$ cells were washed in 1× Binding buffer and incubated with Annexin V-FITC or -APC (1:20) in 1× Binding buffer for 15 min at room temperature. Stained cells were finally resuspended in 1× Binding buffer containing DAPI (1 µg/ml), SYTOX Blue, or PI (1 µg/ml) before analysis. For intracellular Ki-67/DAPI staining, unfractionated BM cells were stained with Lin-PECy5, c-Kit-APC, Sca-1-PECy7, CD48-APCeF780, and CD150-PE, and then fixed and permeabilized with Cytofix/Cytoperm buffer for 20 min on ice. After washing with Perm/Wash (BD Biosciences), cells were stained with anti-Ki-67-FITC (1:100) in Perm/Wash for 30 min on ice, washed with Perm/Wash and then resuspended in Perm/Wash containing 1 µg/ml DAPI before analysis. Cell sorting was performed on FACS Aria IIIu or Fusion (Becton Dickinson). All data were collected on

FACS Aria IIIu, Celesta, or LSRFortessa (Becton Dickinson) and analyzed with FlowJo (BD Biosciences, v10.10.0). For detection of FRET from ECFP to Ypet in the SMART probe, ECFP and FRET signals excited by a 405 mm laser were detected with 450/40 and 530/30 filters, respectively, on a FACS Aria IIIu or LSRFortessa. Cells with an increase in FRET/CFP ratio and a decrease in CFP signals were gated as FRET/CFP$^{hi}$.

### Ex vivo assays
For ex vivo HSC expansion, 500 HSCs were directly sorted per well of a 96-well flat-bottom plate in 200 µL of Ham's F-12 media (Gibco) supplemented with 0.1% polyvinyl alcohol, 10 mM HEPES (Gibco), 1× penicillin–streptomycin–ʟ-glutamine, 1× insulin–transferrin–selenium–ethanolamine (Gibco), 10 ng/mL mouse SCF (BioLegend), and 100 ng/mL mouse TPO (PeproTech) as described previously[69]. After 7 days, cells were collected and counted on Vi-CELL XR, and the frequency of EPCR$^+$ HSCs (CD41$^-$/Mac-1$^-$/FcγR$^-$/c-Kit$^+$/Sca-1$^+$/CD48$^-$/CD150$^+$/EPCR$^+$) was evaluated by flow cytometry. For autophagy flux analyses, HSCs were sorted and incubated with DALGreen (1 µM) for 30 min at 37 °C in 2% FBS/PBS. After being washed twice with 2% FBS/PBS, 1000–2000 stained cells were seeded per well of 96-well flat bottom plate, incubated for 8 h at 37 °C, 5% CO$_2$ in 200 µL of cytokine-free IMDM supplemented with 5% FBS, 1× penicillin–streptomycin–ʟ-glutamine, 0.1 mM non-essential amino acids, 1 mM sodium pyruvate, and 50 µM 2-mercaptoethanol with or without 5 nM bafilomycin A1, and analyzed by flow cytometry.

### Retroviral transduction of HSCs
For transduction of N-terminal FLAG-MLKL, 3×FLAG was added in frame at the N-terminus of mouse *Mlkl* cDNA and cloned into the *pMY-IRES-GFP* retroviral vector. The *pMY-IRES-GFP* and *pMY-3×FLAG-MLKL-IRES-GFP* were transfected into Plat-E packaging cells with FuGENE HD, and the culture supernatants were obtained at 40–48 h and filtered through 0.45 µm PVDF filters. For transduction of the C-terminal truncation mutant of RUNX1, the culture supernatant of 293GPG packaging cells stably transfected with the *pMY-RUNX1S291fs-IRES-GFP* retroviral vector was collected at 48–60 h after tetracycline withdrawal and filtered through a 0.45 µm PVDF filter. For HSC transduction, 2000 WT and *Mlkl*$^{-/-}$ HSCs (CD45.2$^+$) were directly sorted per well of a 96-well flat bottom plate pre-coated with RetroNectin (Takara Bio) in 200 µL SF-O3 media (EIDIA) supplemented with 0.2% BSA, 100 ng/mL mouse SCF, and 100 ng/mL human TPO, incubated overnight at 37 °C, 5% CO$_2$, and subjected to magnetofection for 1 h in the retroviral supernatants with a 96-well magnetic plate and ViroMag R/L. The medium was then replaced with fresh cytokine-containing media, and cells were incubated for 48 h before the re-isolation of live PI$^-$/GFP$^+$ transduced cells for transplantation experiments.

### Immunofluorescence and proximity ligation assay
A total of 2000 HSCs were sorted and pipetted onto MAS adhesive glass slides (Matsunami TF1205M) or µ-Slide 18 Well (ibidi), incubated for 30 min at 4 °C, fixed with 4% paraformaldehyde for 10 min at room temperature, and permeabilized in 0.3% Triton X-100/PBS for 2 min at room temperature. For γH2AX immunofluorescence, slides were blocked in 2% BSA/PBS for 1 h at 4 °C and incubated overnight at 4 °C in 2% BSA/PBS with anti-γH2AX antibody (1:500). Slides were then washed 3 times in PBS and incubated for 1 h at room temperature in 2% BSA/PBS with AF488-conjugated goat anti-mouse IgG (1:200). For FLAG-MLKL immunofluorescence, slides were blocked in 2% BSA/PBS for 1 h at 4 °C and incubated overnight at 4 °C in 2% BSA/PBS with anti-MLKL (1:500) and anti-FLAG (1:500) antibody. Slides were then washed 3 times in PBS and incubated for 1 h at room temperature in 2% BSA/PBS with AF647-conjugated goat anti-mouse IgG (1:500) and AF594-conjugated goat anti-rabbit IgG (1:500). Slides were then washed 3

times in PBS, incubated with 1 mg/ml DAPI/PBS for 5 min, and washed twice in PBS. Slides were finally mounted with coverslips by using ProLong Glass or Gold (Invitrogen). Cells were imaged on an A1Rsi inverted confocal microscope with a ×100 objective (Nikon) or an LSM 980 Airyscan inverted confocal microscope with a ×63 objective (Zeiss). For the proximity ligation assay, slides were blocked in Duolink blocking solution (Sigma-Aldrich) and incubated overnight at 4 °C in Duolink antibody diluent (Sigma-Aldrich) with primary antibodies against COX-IV (1:400) and p-MLKL (1:1600). Slides were washed twice in 1× Duolink wash buffer A (Sigma-Aldrich) and incubated for 1 h at 37 °C in Duolink antibody diluent with PLUS and MINUS probes (1:5; Sigma-Aldrich). Slides were then washed twice in 1× Duolink wash buffer A, incubated for 30 min at 37 °C in 1× Duolink ligation buffer with ligase (1:40, Sigma-Aldrich), washed twice in 1× Duolink wash buffer A, and incubated for 100 min at 37 °C in 1× Duolink amplification buffer with polymerase (1:80; Sigma-Aldrich). After being washed twice in 1× and once in 0.01× Duolink wash buffer B (Sigma-Aldrich), slides were finally mounted with coverslips by using Duolink in situ mounting medium with DAPI (1:80; Sigma-Aldrich), and cells were imaged on an N-SIM super-resolution confocal microscope with a 100× objective (Nikon) or an LSM 980 Airyscan inverted confocal microscope with a ×63 objective (Zeiss). For quantification, cells were randomly captured, and γH2AX or PLA foci were counted by eye using an NIS-Elements Viewer (Nikon, v4.11.0) or Image J (v1.53c).

## Cytokine profiling

For collecting BM fluids, two femurs and two tibiae per mouse were flushed out with the same 200 μL of 2% FBS/PBS by using a 1 mL syringe with a 23 G needle and were spun down at 300×$g$ for 5 min to remove BM cells. Supernatants were further clarified by centrifugation at 12,000×$g$ for 10 min and stored at −80 °C until use. 50 μL of 2× diluted samples were analyzed with a Bio-Plex Pro mouse cytokine 23-Plex panel on a Bio-Plex 200 analyzer (Bio-Rad) according to the manufacturer's protocol. Cytokine concentrations were calculated using standard curves. Morpheus was used to generate heatmaps showing normalized cytokine levels with hierarchical clustering of samples and cytokines with Euclidean distance.

## RNA-seq

RNA was purified from 10,000 HSCs isolated from WT and $Mlkl^{-/-}$ mice with the RNeasy Plus Micro Kit. Double-stranded cDNA was generated using the SMART-Seq HT Kit and fragmented using the M220 Focused ultrasonicator (Covaris). Sequencing libraries were prepared using the NEBNext Ultra DNA Library Prep Kit (New England Biolabs). Different index primers were used for multiplexing samples in one lane, and pooled libraries were sequenced on NextSeq2000 (Illumina) with single-read 70 base pairs. Data quality was verified by FastQC (v0.12.0), and demultiplexing was performed with bcl2fastq (v2.20). Sequencing reads were mapped to the mouse reference genome (mm10) with HISAT2 (v2.2.1) and quantified with StringTie (v2.2.3). Normalization and pairwise differential expression analyses were performed using edgeR (v3.30.3). Principal component analysis was performed using standard packages in R (v4.0.2), and plots were generated using the first 2 principal components. The Benjamini−Hochberg method was used to correct type I errors for multiple gene-level comparisons, with Wald tests applied and a $q$-value cutoff of 0.01 set to define differentially expressed genes (DEGs). Identified DEGs were visualized in volcano plots and area-proportional Venn diagrams using R. Gene set enrichment analysis (v4.3.0) was performed using the 50 "MH" orthology-mapped hallmark gene sets in mouse MSigDB collections, and only those with $P < 0.05$ were displayed as significantly enriched gene sets.

## ATAC-seq

For each biological replicate, 10,000 HSCs were isolated from six young WT mice, six young $Mlkl^{-/-}$ mice, three aged WT mice, and three

$Mlkl^{-/-}$ mice. All HSC replicates were prepared and processed on the same day to avoid potential batch effects. At harvest, cells were lysed and immediately transposed as described previously. In brief, HSCs were resuspended in cold lysis buffer (10 mM Tris−HCl, pH 7.4, 10 mM NaCl, 3 mM MgCl$_2$, and 0.1% IGEPAL CA-630) and incubated for 10 min on ice. Samples were then spun down at 600×$g$ for 10 min at 4 °C and incubated for 35 min at 37 °C in 50 μL of transposase reaction mix consisting of 25 μL Tagment DNA buffer (Illumina), 2.5 μL Tagment DNA enzyme (Illumina), and 22.5 μL nuclease-free water. Transposed fragments were purified with MinElute PCR Purification Kit (QIAGEN), eluted in 14 μL of nuclease-free water, and stored at −20 °C until library preparation. After the optimization of PCR cycle number, libraries were semi-quantified with SYBER Green I Nucleic Acid gel Stain (Takara Bio) to optimize the number of PCR cycles and then amplified using NEBNext High Fidelity 2×PCR Master mix (New England Biolabs) and index primers. Amplified libraries were purified with the MinElute PCR Purification Kit, size-selected between 240 and 360 base pairs by using BluePippin (Sage Science), and sequenced with NextSeq500 (Illumina) with 70 bp single-read length. Sequencing reads were mapped to the mouse reference genome (mm10) with Bowtie2 (v2.5.3). Peaks were called with MACS2 (v2.2.7.1) using the nomodel function, with a $q$-value cutoff of 0.001 set to define accessible peaks in each sample. The catalog of all peaks called in any samples was generated by merging all called peaks that overlapped by at least one base pair by using the bedtools (v2.31.0) merge function. As a result, a total of 76,286 merged peaks were detected and used as a map file for downstream processing. Reads at each peak in the catalog were quantified with the bedtools map function by using bed files of each sample. Differentially accessible regions (DARs) were detected using edgeR with a $q$-value cutoff of 0.01 and visualized as volcano plots and area-proportional Venn diagrams using R.

## Transmission electron microscopy

A total of 100,000 HSCs were sorted in 2% FBS/PBS, spun down at 300×$g$ for 5 min at 4 °C, and fixed for 2 h at room temperature with 1% glutaraldehyde in 0.1 M sodium phosphate buffer pH 7.4. After fixation, cells were rinsed and post-fixed on ice for 2 h with 2% osmium tetroxide in 0.1 M sodium phosphate buffer, pH 7.4. Cells were then washed, dehydrated in a graded series of ethanol, and embedded in Epon 812 resin mixture (TAAB). Semi-thin sections of about 0.7 μm thickness were cut on an EM UC7 ultramicrotome (Leica), stained with 0.2% toluidine blue, and examined under an ECLIPSE Si microscope (Nikon). Ultra-thin sections were cut, stained for 5 min at room temperature with uranyl acetate and lead citrate, and imaged on a JEM-1400Flash electron microscope (JEOL). Mitochondria and autophagosomes were counted by eye, and mitochondrial morphology was analyzed with ImageJ.

## Immunoelectron microscopy

A total of 100,000 HSCs were sorted in 2%FBS/PBS, spun down at 300×$g$ for 5 min at 4 °C, and fixed for 30 min on ice with 4% paraformaldehyde and 0.2% glutaraldehyde in 0.1 M sodium phosphate buffer pH 7.4. After fixation, cells were rinsed, dehydrated in a graded series of ethanol, and embedded in LR white medium-grade resin (Electron Microscopy Sciences). Semi-thin sections of about 0.7 μm thickness were cut on an EM UC7 ultramicrotome, stained with 0.2% toluidine blue, and examined under an ECLIPSE Si microscope. Ultra-thin sections were cut and blocked with 1% BSA/PBS for 10 min at room temperature. The sections were then incubated for 1 h at room temperature in 1% BSA/PBS with a primary antibody against p-MLKL (1.0 μg/mL). Sections were washed 3 times in PBS and incubated for 1 h at room temperature in 5 nm gold-conjugated secondary antibody against rabbit IgG (1×). Sections were washed 3 times in water, post-fixed in 1% glutaraldehyde for 10 min, and stained with 1% uranyl acetate for 3 min at room

temperature. Cells were imaged on a JEM-1400Flash electron microscope (JEOL).

## Recombinant protein expression and purification
N-terminal four-helix bundle domain of recombinant mouse MLKL protein (residues 1–169; MLKL-NTD), N-terminal FLAG-tagged MLKL-NTD (FLAG-MLKL-NTD), and C-terminal pseudokinase domain of recombinant mouse MLKL protein (residues 179–464; MLKL-CTD) were expressed and purified from BL21 (DE3)-RIPL *Escherichia coli* strains. Briefly, cells were grown in 2XTY medium (20 g/L tryptone, 10 g/L yeast extract, 5 g/L NaCl, 50 μg/ml kanamycin, 30 μg/ml chloramphenicol) to an optical density at 600 nm of 0.8 and induced with 0.5 mM isopropylthio-β-galactoside for 16 h at 22 °C. Pellets were resuspended in buffer (25 mM Tris–HCl pH 7.4, 500 mM NaCl, 1 mM TCEP, 10% glycerol, 0.05% Triton-X 100, 25 mM imidazole), lysed at high pressure, clarified by centrifuge, subjected to $Ni^{2+}$ column, washed and eluted with buffer (25 mM Tris–HCl pH 7.4, 500 mM NaCl, 1 mM TCEP, 10% glycerol, 0.05% Triton-X 100, 250 mM imidazole). Elution was collected and incubated with TEV protease at 4 °C overnight to cleave off the His tag, followed by a reverse $Ni^{2+}$ column to remove the TEV protease and any undigested His tag protein. The digested proteins were further polished by Superdex S200 gel filtration column pre-equilibrated with gel filtration buffer (20 mM Tris pH 7.4, 200 mM NaCl, 1 mM TCEP, 1 mM EDTA). Selected fractions were pooled and concentrated by centrifugal ultrafiltration to 1 mg/ml and dialyzed in the buffer (10 mM Tris–HCl pH 7.5, 10 mM KCl, 250 mM sucrose, 1.5 mM $MgCl_2$).

## Isolation and treatment of mitochondria
Mitochondria were isolated from mouse liver tissue using the Mitochondria Isolation Kit for Cultured Cells (Abcam) according to the manufacturer's protocol. Briefly, ~50 mg of liver tissue was dissected, immediately transferred into 500 μL of Reagent A pre-dispensed in the Dounce homogenizer tube, and homogenized by 30 strokes with pestle A. Homogenate was centrifuged at 1000×*g* for 10 min at room temperature, and the supernatant (SN1) was collected. The pellet was then resuspended in 500 μL of Reagent B, homogenized by pipetting up and down 30 times using a P1000 micropipette, and the supernatant was collected (SN2) after centrifugation at 1000×*g* for 10 min at room temperature. SN1 and SN2 were combined, and mitochondrial fractions were pelleted by centrifugation at 12,000×*g* for 15 min at room temperature. The mitochondrial pellet was resuspended in 200 μL of mitochondrial buffer (10 mM Tris–HCl pH 7.5, 10 mM KCl, 250 mM sucrose, 1.5 mM $MgCl_2$), and 2.5 μL of mitochondria were incubated in 87.5 μL of mitochondrial buffer for 45 min at 37 °C with 20 μM recombinant MLKL-NTD, 20 μM recombinant FLAG-MLKL-NTD, 20 μM recombinant MLKL-CTD, or 50 μM CCCP (Thermo Fisher). Treated mitochondria were stained in mitochondrial buffer containing 500 nM TMRM (Invitrogen) and 200 nM MitoTracker Green (Thermo Fisher) for 30 min at 37 °C before flow cytometry analysis. All procedures were performed without vortexing to avoid mechanical disruption of mitochondria and preserve mitochondrial membrane potential.

## Seahorse assays
A total of 50,000–100,000 HSCs were sorted in SF-O3 and spun down at 300×*g* for 5 min at 4 °C. Cells were resuspended in Seahorse XF basic DMEM (Agilent) supplemented with 10 mM glucose, 1 mM pyruvate, and 2 mM L-glutamine, seeded in a cell-culture plate pre-coated with Cell-Tak (Corning) at a density of 50,000–70,000 cells per well, and spun down by centrifugation at 400×*g* for 5 min at room temperature. Oxygen consumption rate and extracellular acidification rate were measured at baseline and after sequential injections of 1 μM oligomycin, 2 μM FCCP, 0.5 μM rotenone plus 0.5 μM antimycin, and 50 mM 2-deoxy-D-glucose on a Seahorse XFe96 Extracellular Flux Analyzer (Agilent) according to the manufacturer's instructions as described previously[70]. Data were normalized to seeded cell numbers, analyzed with Seahorse Analytics (v1.0.0-520), and exported to GraphPad Prism (v10.4.0).

## Quantification and statistical analysis
Data are represented as mean ± standard error (s.e.m.) unless otherwise specified in the figure legends. Statistical analysis was performed using R for RNA-seq and ATAC-seq data and Prism for all the other data. The statistical test used, the number of biological replicates, and the definition of biological replicates are indicated in the figure legends. Briefly, two-tailed Student's *t*-test and Mann–Whitney *U*-test were used when two groups were compared, and one-way ANOVA, two-way ANOVA, Welch and Brown–Forsythe test, and Kruskal–Wallis test were used when three or more groups were compared. For multiple comparisons, *p*-values were adjusted either by Šídák correction, Fisher's least significant difference (LSD) test, or the false discovery rate of 5% based on the two-stage linear step-up procedure of Benjamini, Krieger, and Yekutieli, and adjusted *p*-values were subsequently used to indicate significance. For transplantation experiments, outliers were excluded before statistical analyses based on the PB donor chimerism at 4 months post-transplantation by using the ROUT method with a *Q* value set to 1%. Recipient mice with <1% PB donor chimerism at 4 months post-transplantation were excluded from the analyses of donor lineage distribution. Survival rates were compared using the Mantel–Cox log-rank test. Statistical significance is reported using exact *P* values.

## Reporting summary
Further information on research design is available in the Nature Portfolio Reporting Summary linked to this article.

## Data availability
Bulk RNA-seq data and ATAC-seq data have been deposited at Gene Expression Omnibus and are publicly available under the accession number GSE285111. Source data are provided with this paper.

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

## Acknowledgements

We thank Manolis Pasparakis for *Mlkl*⁻/⁻ mice, Avi Ashkenazi for recombinant mouse TNFα, Yosuke Tanaka for Plat-E cells, and Goro Sashida for the 293GPG cells expressing *pMY-RUNX1S291fs-IRES-GFP*. We are grateful to Arthur Flohr Svendsen for the Neo-1-staining protocol, and to Richard Heath and Wei Wang for recombinant mouse MLKL variants. We thank Yumiko Ishii, David Cullins, and Emilia Kooienga for flow cytometry support, and Aaron Taylor and Aaron Pitre for assistance with confocal microscopy. We acknowledge Kazusa DNA Research Institute for RNA and ATAC sequencing, Motohiko Oshima for bioinformatics support, and Makiko Miyota, Fumiko Maki, Rei Imai, and Naomi Otis for mouse husbandry assistance. We thank Rebekah Doerfler for scientific editing and members of the Yamashita and Iwama laboratories for helpful discussions. Y.Y. was supported by a JSPS research fellowship. This work was supported by JSPS KAKENHI (23H02707) and the Takeda Science Foundation (H.N.); JSPS KAKENHI (24H00640, 25H01435, 25K22629) and AMED (JP20gm1210011, 25bm1123063) (K.T.); and ALSAC, JSPS KAKENHI (20K17395, 22H03101), the Uehara Memorial Foundation, and Kato Memorial Bioscience Foundation (M.Y.). Shared resource core facility support was provided by ALSAC and NIH grant P30CA21765.

## Author contributions

M.Y. conceived and directed the project. Y.Y. and M.Y. designed the experiments. Y.Y. J.Y., and M.Y. performed the experiments and analyzed the data. A.S.-T. prepared plasmids and retroviral supernatants. Y.Y. and S.K. prepared RNA-seq and ATAC-seq libraries. K.S. and H.N. generated SMART-Tg mice. S.M. and H.N. prepared aged WT and *Ripk3*⁻/⁻ SMART-Tg mice. Y.W. and H.S. prepared the electron microscopy specimens and acquired the images. Y.S. and K.T. performed the Seahorse assay. Y.F. analyzed the data. H.S., H.N., K.T. and A.I. discussed the results. Y.Y. and M.Y. wrote and edited the manuscript. All the authors contributed to reading and editing the manuscript.

## Competing interests

The authors declare no competing interests.
