## [Transparent Peer Review file · Nature Communications]

Non-necroptotic MLKL function damages mitochondria and promotes hematopoietic stem cell aging

Corresponding Author: Dr Masayuki Yamashita

Version 0:

Reviewer comments:

Reviewer #1

(Remarks to the Author)

In their study, Yamada et al. investigated the role of the necroptosis factor MLKL in HSC aging. Using SMART-Tg mice they demonstrate an increase in MLKL, measured through FRET, upon treatment with inflammation-inducing agents (poly-IC, LPS and TNF-alpha), as well as following replication stress and in aged mice. This increase in MLKL was correlated with poorer HSC reconstitution, which was also observed in aged mice. Utilizing an MLKL knockout strain, the authors showed that MLKL deficiency partially rescued the phenotype of poorer HSC reconstitution and other aging-associated features. Furthermore, Yamada et al. discovered that activated MLKL accumulated in mitochondria leading to mitochondrial dysfunction in aged HSCs, which could partially be corrected by MLKL deficiency.

The study is very comprehensive and well-executed, presenting intriguing findings about the potential role of MLKL beyond necroptosis. However, my concern is that the cell death-independent role of MLKL is based solely on Figure 2B, which includes only four mice from two experiments, showing quite some variation, especially in the WT pIC group. I recommend including more replicates in Figure 2B, and incorporating cell death analysis as validation in subsequent figures, such as in the Figure 2F experiment. Another concern is that the authors have not conclusively shown that MLKL underlies aging, as stated in the title and throughout the manuscript. While it is evident that MLKL contributes to or even promotes aging, I disagree to the statement that MLKL is the driver of aging considering that the authors only demonstrate partial restoration when MLKL is knocked out (e.g. Figure 4G, Figure 6F).

Further points for clarification:

1. Line 88 states that necroptosis is selectively and transiently activated in HSCs. However, the data clearly show activation in both HSCs and multipotent progenitors upon treatment with pIC, LPS and TNF α (Figure 1B). Was lymphoid progenitors analyzed as well?
2. In Figure 1D, FRET high and low cells were sorted and transplanted. It would be informative to show the sorting gates for FRET high and low. Does FRET low indicate "negative" cells or truly low cells? Would the truly FRET negative cell not be a better control than PBS control?
3. In transplantation experiments it would be informative to see the total reconstituted levels of each blood lineage (i.e. out of total blood cells) rather than only relative reconstitution frequencies within the donor-derived compartment. This is especially relevant if total donor reconstitution levels vary a lot between recipient mice.
4. Could the absence of significant transcriptional differences between WT and MLKL $^{-/-}$ be due to the bulk RNA-seq mode, which might omit differences that would otherwise be detected with single cell RNA-seq?
5. Why were young mice not included in Figure 6E, G, H for comparison?
6. Do MLKL $^{-/-}$ mice live longer?
7. Have you checked for any differences between females and males in inflammation-induced MLKL activation and in aging?

Reviewer #2

(Remarks to the Author)

The work by Yamada et al. investigates a cell death-independent role of MLKL in HSC inflammation and aging, which does

not regulate the transcriptome; instead, it impairs mitochondrial fitness. This work is well-conducted and controlled; the phenotype is robust; and the scientific rigor is high. One of several experiments I appreciate a lot is the N-flag MLKL rescue one. As N-flag MLKL cannot induce necroptosis but can still rescue the MLKL KO HSC phenotype, I believe the "cell-death-independent role" of MLKL in HSC is very solid. Although the overall quality of this work is high, some minor concerns can be addressed to enhance its excellence further.

1 As the authors do have the RIP3 KO mice, it would be great to test whether RIP3 KO can phenocopy MLKL KO, in assays such as Fig. 2d, 3d and 4g. If the aged RIP3 KO mice are not available, then just discuss this as a limitation of this study, as I don't want to delay the publication for too long. This RIP3 KO mice work will complete the story and add more impact to this study.

2 For the N-flag-MLKL rescue experiments, it would be interesting to know whether N-flag-MLKL can still target the mito (via PLA) or impair mito function. This is important because N-flag-MLKL can still localize to the plasma membrane but cannot perform pore formation on the plasma membrane (<https://www.nature.com/articles/cdd201570>). So this work would be informative.

3 For the title, since it is not "sub-lethal", it is "necroptosis-independent", please revise it to align with the current abstract. And I would also suggest that include "inflammation" on the title as I think this work is more than just studying "aging".

Reviewer #3

(Remarks to the Author)

The manuscript by Yamada et al. describes the role of Mkl in hematopoietic stem cells (HSC). Authors showed that Mkl is activated in HSC and multipotent progenitor (MPP) pool upon inflammation using FRET based sensor that monitor MLKL translocation to membrane. However, they showed that this activation does not lead to cell death of HSCs, as expected by the known function of Mkl in necroptosis. Next, authors performed series of experiments on Mkl^{-/-} mice, and demonstrated that such mice do not show many established hallmarks of HSC aging or strong inflammation-driven exhaustion of HSC function. Finally, the authors showed the link of Mkl with age-related changes in HSCs and demonstrated that Mkl^{-/-} HSCs has more ATP production and sustained glycolytic flux.

Significance and innovation:

In the field of HSC aging there are many phenotypes reported that show premature aging of HSC. However, there very few studies demonstrating phenotypes that protect HSCs from acquiring aging hallmarks, like myeloid-biased or stem cell pool expansion with reduced reconstitution potential. Therefore, in my opinion the presented work is significant contribution to the field. While I am not an expert in the field of programmed cell death and necroptosis, the observation of the Mkl translocation to membrane without induction of necroptosis seem to be not well studied, with some prior papers (eg. Vucur et al Immunity 2023) cited.

My major question/remarks:

First is the conclusion about "sublethal necroptosis" stated within the title and text. While the term "sublethal necroptosis" has been used in field previously, in my opinion it might be misleading – either there is necroptosis, and the cell is dying or there is just necroptosis-independent role of Mkl. The fact that the used FRET sensor shows Mkl translocation to the membranes may not necessarily means it is involved in membrane rupture. Now, given the novel link with mitochondrial energy production this together may indicate an unrecognized function of Mkl, that is not related to necroptosis. eg. membrane trafficking. Therefore, I am not fully convinced with the term sublethal necroptosis.

Second, I think it is not logically correct to state that "Sublethal necroptosis underlies hematopoietic stem cell aging" based on the observation of Mkl deficiency result is resistance to age-related changes in HSC. I think, based on what is presented I would state that loss of MLKL protects against hallmarks of HSC aging, what is still an important observation.

The article open next important questions. How the Mkl affects mitochondria, what is its role here? However I fully understand that this a wide research question and may be beyond the current study and was also stated by the authors in the discussion. But one thing I would suggest for consideration is to add that at this point the observed reversal of mitochondrial function may be the reason or the effect of the aging protection.

Finally, I find very interesting that the protection of aging in Mkl^{-/-} mice is not linked to reversal of expansion of the expansion of myeloid-biased HSC defined as CD41⁺. Now, given the authors transplanted the whole pool of HSCs that means that either Mkl-deficiency protects the "balanced" fraction of HSCs during aging or reverse the myeloid and aging phenotype of "myeloid-biased" HSCs. Given it is possible now to prospectively transplant myeloid-biased HSCs (work by Ross et al. using Neo-1 as marker, Nature 2024, Weissman lab) would it be possible to answer this question?

Research strategy:

In my opinion all HSC assays are well and robustly performed, well-presented and I have no doubts about the clear and significant effect of the Mkl-deficiency on HSC aging protection. There are few places where I would suggest some more precise description (below).

Other comments:

- Extended Fig 1 C – the quality of the blot is poor, not sure if it is needed to include it – maybe the IHC is more informative
- Authors wrote observed lower engraftment potential and a tendency toward reduced lymphopoietic potential of FRET/CFPhi HSCs compared to FRET/CFPhi HSCs. (Fig 1d-f) But this mostly B lineage at the cost of myelo, T cells are not changed here.

- In pIC experiments authors stated that Mkl1 deficiency significantly sustained the lymphoid potential, but to be precise is still only B lineage, and T lin even decreased.
- Fig. 1 – it is stated that Mkl1 activation is preferential in HSC, but the data shows it is also observed in MPPs to the same extend.
- The same is in the Fig. 3a – statement that the activation is preferential to HSCs would require comparison to more populations (eg. MPPs) not only to MP
- Authors discussed the changes of the mitochondria with aging as “HSC memory”. While I found the concept of HSC memory very valid and interesting, in my opinion the term memory makes sense when the memory govern some advantage to secondary stimuli/challenge. Otherwise for me it is just another aging hallmark/feature. I can't see what kind of “memory advantage” would the observed decline in mitochondria decline would provide.

Version 1:

Reviewer comments:

Reviewer #1

(Remarks to the Author)

In the revised study by Yamada et al., the authors have comprehensively addressed the concerns raised during the previous review round by incorporating additional experiments and new analyses. These additions substantially strengthen the manuscript and enhance the clarity of its conclusions. Notably, the inclusion of experiments investigating MLKL in mitochondrial function and the role of RIPK3–MLKL in age-related HSC functional decline, through the analysis of NEO-positive and NEO-negative HSCs, represents a significant improvement.

Overall, the revisions have increased the quality and impact of the study. I am satisfied with the authors' responses and the new data presented.

Reviewer #2

(Remarks to the Author)

Accept

Reviewer #3

(Remarks to the Author)

The authors have addressed all of my comments. I find the observation of reversed Neo-1 expression and the analysis of myeloid-biased HSCs following pIC treatment particularly interesting and representing an added value to the manuscript. I also acknowledge that the authors performed Neo-1⁺/Neo-1⁻ HSC transplantation experiments, which support the conclusion that RIPK3–MLKL blockade acts on the Neo-1 fraction; however, the number of mice included in these experiments is relatively limited.

Nevertheless, in my opinion, the revision is comprehensive and has significantly improved the manuscript.

Point-by-point responses to reviewers' comments:

We wish to start by expressing our sincere appreciation to the reviewers for their constructive comments, which have helped us significantly improve our manuscript. The revision experiments have led us to discover two major findings: (1) the RIPK3-MLKL axis promotes age-related HSC functional decline primarily via depleting NEO-1⁻ lineage-balanced functional HSCs, and (2) active MLKL directly impairs mitochondrial membrane potential via mechanisms dependent on and independent of its pore-forming function. We have also revised the title and text in accordance with the reviewers' suggestions as well as the revised conclusions. Below, we provide a point-by-point response to each of the reviewer's comments.

Reviewer #1

We thank the reviewer for her/his pertinent feedback on our manuscript. We are encouraged by the reviewer's recognition of the scope and rigor of our study, and her/his acknowledgment of the potential role of MLKL beyond necroptosis. The updated results now more convincingly support cell death-independent role of the RIPK3-MLKL axis in age-related HSC functional decline. We have changed the title and revised the text in accordance with the reviewer's suggestion along with the new experimental data. We have also addressed the other concerns raised by the reviewer by performing additional experiments and reanalyzing existing data.

Comment 1. *In their study, Yamada et al. investigated the role of the necroptosis factor MLKL in HSC aging. Using SMART-Tg mice they demonstrate an increase in MLKL, measured through FRET, upon treatment with inflammation-inducing agents (poly-IC, LPS and TNF-alpha), as well as following replication stress and in aged mice. This increase in MLKL was correlated with poorer HSC reconstitution, which was also observed in aged mice. Utilizing an MLKL knockout strain, the authors showed that MLKL deficiency partially rescued the phenotype of poorer HSC reconstitution and other aging-associated features. Furthermore, Yamada et al. discovered that activated MLKL accumulated in mitochondria leading to mitochondrial dysfunction in aged HSCs, which could partially be corrected by MLKL deficiency. The study is very comprehensive and well-executed, presenting intriguing findings about the potential role of MLKL beyond necroptosis. However, my concern is that the cell death-independent role of MLKL is based solely on Figure 2B, which includes only four mice from two experiments, showing quite some variation, especially in the WT pIC group. I recommend including more replicates in Figure 2B, and incorporating cell death analysis as validation in subsequent figures, such as in the Figure 2F experiment.*

Response. We have added additional experimental mice to the results shown in Fig. 2b, which now include 7 biological replicates from 3 independent experiments and support our initial conclusion that MLKL activation does not mediate HSC death (Fig. 2b). We have also confirmed that overexpression of N-terminal FLAG-tagged MLKL does not increase HSC death (Fig. 2i) but decreases their regenerative potential with additional experiments (Fig. 2j-l) To assess the impact of 5-FU-induced activation of the RIPK3-MLKL axis on HSC death, we utilized MLKL-SA2 knock-in mice (JAX #039024) where MLKL cannot be activated by RIPK3-dependent phosphorylation due to the two serine-to-alanine substitutions at codons 345 and 347 of the *Mkl1* locus (Yeap HW et al. EMBO J 2025, PMID: 40128366). Due to the substantial reduction of c-Kit⁺ expression on functional HSCs early after 5-FU administration, we omitted c-Kit from our analysis panel and instead added EPCR to enrich HSC populations as described previously (Umemoto T et al. J Exp Med 2018, PMID: 29946000). While MLKL activation was observed in HSCs at 16 h post-5-FU injection, cell death frequencies in Lin⁻/CD48⁻/CD150⁺/Sca-1⁺/EPCR⁺ HSCs were kept as low as ~5% at that time (Fig. 3c, Supplementary Fig. 4d). Moreover, even though the HSC death

frequencies elevated up to ~35% at 48 h post-5-FU administration, inactivation of the RIPK3-MLKL axis had no significant impact on the 5-FU-induced HSC death (Fig. 3c, Supplementary Fig. 4d), confirming that 5-FU-induced MLKL activation does not cause HSC death.

Comment 2. *Another concern is that the authors have not conclusively shown that MLKL underlies aging, as stated in the title and throughout the manuscript. While it is evident that MLKL contributes to or even promotes aging, I disagree to the statement that MLKL is the driver of aging considering that the authors only demonstrate partial restoration when MLKL is knocked out (e.g. Figure 4G, Figure 6F).*

Response. We apologize for any unintended consequences arising from the manuscript's title and wording, which may have been misleading. We totally agree that the partial restoration of aging hallmarks in *MLKL*^{-/-} HSCs rather indicates the RIPK3-MLKL axis is one of the mechanisms that mediate HSC aging, and this was indeed the conclusion we highlighted in the Discussion section of the original manuscript. As detailed in our response to reviewer #2, comment 3, we have performed additional experiments using isolated mitochondria and recombinant MLKL protein, and demonstrated that MLKL directly impairs mitochondrial function (Fig. 6h). To better reflect these notions, we have now changed the title to “non-necroptotic MLKL function damages mitochondria and promotes hematopoietic stem cell aging” and replaced the term “driver” by “mediator” throughout the manuscript.

Comment 3. *Line 88 states that necroptosis is selectively and transiently activated in HSCs. However, the data clearly show activation in both HSCs and multipotent progenitors upon treatment with pIC, LPS and TNF α (Figure 1B). Was lymphoid progenitors analyzed as well?*

Response. We agree that MPP2 and MPP3/4 also showed FRET activation and have included this point in the text (lines 75–76). Regarding the FRET analysis on CLP, the key marker IL-7 α could not be included in the original flow panel designed for identification of HSCs and MPPs and used in Fig. 1b, due to the limited numbers of fluorescent channels available after allotting three colors required for the FRET analysis (i.e., CFP, FRET, YPet). Thus, we have performed additional experiments with the new flow panel focusing on the CLP and demonstrated that while the FRET signal was significantly increased in Lin⁻/Sca-1⁺/c-Kit⁺ cells that consist of HSCs and MPPs, no such increase was observed in CLPs (Supplementary Fig. 1c). Similarly, FRET signals were selectively elevated in HSCs/MPPs but not in CLPs or MPs upon 5-FU treatment (Fig. 3b, Supplementary Fig. 4a). These results consolidate our conclusion that the RIPK3-MLKL axis is preferentially activated in HSCs and MPPs.

Comment 4. *In Figure 1D, FRET high and low cells were sorted and transplanted. It would be informative to show the sorting gates for FRET high and low. Does FRET low indicate “negative” cells or truly low cells? Would the truly FRET negative cell not be a better control than PBS control?*

Response. We have now explicitly shown the sorting gates to isolate LPS-treated FRET/CFP^{hi} and FRET/CFP^{lo} HSCs for transplantation studies (Fig. 1e). Using the FRET signals in PBS-treated HSCs, we first defined the gates to select cells with the top 10% of FRET/CFP ratio as FRET/CFP^{hi} (which normally corresponds to cells showing decrease in CFP intensity due to FRET) and those with the bottom 30% as FRET/CFP^{lo} and then applied the same gates to LPS-treated HSCs. As a result, we selected cells with the bottom 4% of FRET/CFP ratio as FRET/CFP^{lo} in LPS-treated HSCs, which we believe selects truly FRET-low cells. As the MLKL activation status is considered the sole difference between LPS-treated FRET/CFP^{hi} and FRET/CFP^{lo} HSCs, we indeed used FRET/CFP^{lo} HSCs as the control to dissect the effect of LPS-induced MLKL activation on HSCs as explained in the original manuscript (lines 84–86).

However, the relatively similar engraftment levels between PBS-treated HSCs and LPS-treated FRET/CFP^{lo} HSCs (Fig. 1f) also indicates that LPS-treated HSCs without apparent MLKL activation maintains similar levels of engraftment potential compared to the steady-state HSCs, which we think would be also informative to the readers and thus be kept in the revised manuscript.

Comment 5. *In transplantation experiments it would be informative to see the total reconstituted levels of each blood lineage (i.e. out of total blood cells) rather than only relative reconstitution frequencies within the donor-derived compartment. This is especially relevant if total donor reconstitution levels vary a lot between recipient mice.*

Response. We have reanalyzed the existing transplantation data to show donor chimerism within each lineage of peripheral blood leukocytes. The results were consistent with our conclusions, showing that MLKL deficiency improved the reconstitution levels of lymphoid cell lineages by pIC-treated (Supplementary Fig. 2g), 5-FU-treated (Supplementary Fig. 4i), serially transplanted (Supplementary Fig. 5b), and aged HSCs (Supplementary Fig. 6g).

Comment 6. *Could the absence of significant transcriptional differences between WT and MLKL^{-/-} be due to the bulk RNA-seq mode, which might omit differences that would otherwise be detected with single cell RNA-seq?*

Response. As the reviewer pointed out, we cannot formally exclude the possibility that MLKL might affect a specific HSC subpopulation that could not be detected by bulk RNA-seq. Unfortunately, due to our recent move to St. Jude, we no longer have access to aged *Mkl1^{-/-}* mice to perform single cell RNA-seq. However, in response to the Reviewer #3 comment 4, we have found that blockade of the RIPK3-MLKL axis largely prevents pIC-induced conversion from neogenin-1-negative (NEO-1⁻) lineage-balanced HSCs to NEO-1⁺ myeloid-biased HSCs (Fig. 2d-e). While we could not detect *Neo-1* gene expression changes between WT and *Mkl1^{-/-}* aged HSCs in the bulk RNA-seq, the results support the notion that activation of the RIPK3-MLKL axis shifts NEO-1⁻ lineage-balanced HSCs toward NEO-1⁺ myeloid-biased HSCs. Thus, we have ameliorated the corresponding statement (lines 199–200) and discussed the limitation of our bulk approaches in the Discussion section (line 285–286).

Comment 7. *Why were young mice not included in Figure 6E, G, H for comparison?*

Response. Fig. 6d (i.e., Fig. 6e of the original manuscript) has now included quantitative data and statistical analyses for young and aged *Mkl1^{-/-}* HSCs. For Fig. 6f,g (i.e., Fig. 6g,h of the original manuscript), the Seahorse assay requires ~280,000 HSCs (i.e., at least 70,000 cells per well, with a minimum of four biological replicates for reliable measurement). While we could obtain ~100,000 HSCs from single aged mice due to their age-dependent expansion, it is extremely challenging to obtain enough numbers of HSCs from young mice, as we could typically obtain 5,000 HSCs per mouse and would require >50 mice per experiment. We have now clarified this limitation in the Result section (lines 220–221).

Comment 8. *Do MLKL^{-/-} mice live longer?*

Response. Although we did not formally analyze the lifespan of *Mkl1^{-/-}* mice, we did not recognize any apparent difference in terms of their morbidity and mortality, in line with the prior observation (Tovey Crutchfield EC et al. Cell Death Differ 2023, PMID: 36755069). We have included this point in the Discussion section (lines 293–294).

Comment 9. *Have you checked for any differences between females and males in inflammation-induced MLKL activation and in aging?*

Response. We have reanalyzed the existing data of SMART-Tg mice to assess potential sex differences in MLKL activation. The results indicate that inflammation-induced (Supplementary Fig. 1e) and age-related MLKL activation (Supplementary Fig. 6a) was generally conserved between male and female mice, arguing for sex-independent MLKL activation in HSCs.

Reviewer #2

We greatly appreciate the reviewer's positive and thoughtful evaluation of our study. We are pleased that the reviewer values the robustness of our findings and the scientific rigor of our approach, in particular the N-terminal FLAG-tagged MLKL (FLAG-MLKL) rescue experiments. The reviewer raised important questions as to whether RIPK3 acts upstream of MLKL to promote HSC aging, and the mechanism whereby MLKL impairs mitochondrial function in HSCs. We have performed a series of experiments to address these questions and revised the title to better align with the conclusions.

Comment 1. *The work by Yamada et al. investigates a cell death-independent role of MLKL in HSC inflammation and aging, which does not regulate the transcriptome; instead, it impairs mitochondrial fitness. This work is well-conducted and controlled; the phenotype is robust; and the scientific rigor is high. One of several experiments I appreciate a lot is the N-flag MLKL rescue one. As N-flag MLKL cannot induce necroptosis but can still rescue the MLKL KO HSC phenotype, I believe the "cell-death-independent role" of MLKL in HSC is very solid. Although the overall quality of this work is high, some minor concerns can be addressed to enhance its excellence further.*

Response. As described above, we have performed additional experiments to address the concern and strengthen our work. The detailed responses are shown below.

Comment 2. *As the authors do have the RIP3 KO mice, it would be great to test whether RIP3 KO can phenocopy MLKL KO, in assays such as Fig. 2d, 3d and 4g. If the aged RIP3 KO mice are not available, then just discuss this as a limitation of this study, as I don't want to delay the publication for too long. This RIP3 KO mice work will complete the story and add more impact to this study.*

Response. We agree that testing whether RIPK3 is the upstream regulator of MLKL in the observed HSC changes would further strengthen our work. Unfortunately, our recent relocation from Japan to the US prevented our access to the *RIPK3*^{-/-} mice. To circumvent this issue, we have instead taken advantage of UH15-38, a RIPK3 inhibitor recently developed for *in vivo* use (Gautam A et al. Nature 2024, PMID: 38600381). We found that administration of UH15-38 prior to pIC injection showed trend towards ameliorating pIC-induced decline in HSC regenerative capacity and lymphoid differentiation potential (Supplementary Fig. 3a-d), which is reminiscent of the phenotype observed in *Mkl*^{-/-} HSCs. To further address the role of RIPK3-dependent MLKL activation in HSC changes, we utilized MLKL-SA2 knock-in mice (JAX #039024), where MLKL cannot be activated by RIPK3-dependent phosphorylation due to the two serine-to-alanine substitutions at codons 345 and 347 of the *Mkl* locus (Yeap HW et al. EMBO J 2025, PMID: 40128366). We found that MLKL-SA2 and MLKL deficiency similarly attenuate pIC-induced upregulation of age-related marker NEO-1 on HSCs (Fig. 2d-e). These data, together with the data shown in Fig. 2f, further consolidate our conclusion that the RIPK3-MLKL axis promotes age-related functional decline in HSCs. As the inhibitor is not suitable for long-term inhibition of RIPK3 *in vivo*, we

could not apply this approach to other aging model such as sequential 5-FU treatment or organismal aging, and the limitation of our study regarding RIPK3 requirement has been discussed in the Discussion section (lines 267–270).

Comment 3. *For the N-flag-MLKL rescue experiments, it would be interesting to know whether N-flag-MLKL can still target the mito (via PLA) or impair mito function. This is important because N-flag-MLKL can still localize to the plasma membrane but cannot perform pore formation on the plasma membrane (<https://www.nature.com/articles/cdd201570>). So this work would be informative.*

Response. We thank the reviewer for this insightful suggestion that gave us an opportunity to gain mechanistic insights as to how MLKL impairs mitochondrial function. We first confirmed the expression of FLAG-MLKL by immunofluorescence (Supplementary Fig. 3m). We then performed proximity ligation assay (PLA) with p-MLKL and COX-IV antibodies to examine whether FLAG-MLKL can still target mitochondria. The results have convincingly shown induction of PLA foci in FLAG-MLKL-transduced HSCs, supporting mitochondrial localization of N-FLAG-MLKL (Supplementary Fig. 8b). To determine whether MLKL directly impairs mitochondrial membrane potential, mitochondria were isolated from mouse liver and incubated *in vitro* with either (i) N-terminal four-helix bundle domain of recombinant mouse MLKL protein (MLKL-NTD) that oligomerizes and forms pores on the cell membrane without stimulation, (ii) N-terminal FLAG-tagged MLKL-NTD (FLAG-MLKL-NTD) that retains oligomerization capacity but cannot perform pore formation on the cell membrane, or (iii) C-terminal pseudokinase domain of recombinant mouse MLKL protein (MLKL-CTD) that does not oligomerize or form pores on the cell membrane (Hildebrand JM et al. PNAS 2014, PMID: 25288762). Mitochondria were then stained with Tetramethylrhodamine methyl ester (TMRM) as well as MitoTracker green, and TMRM signals within MitoTracker-positive population were analyzed by flow cytometry. TMRM signals in mitochondria were significantly reduced by incubation with MLKL-NTD, and to a lesser extent by FLAG-MLKL-NTD, but not by MLKL-CTD (Fig. 6h). These results indicate that activated MLKL directly impairs mitochondrial membrane potential via mechanisms both dependent and independent of its pore-forming capacity.

Comment 4. *For the title, since it is not "sub-lethal", it is "necroptosis-independent", please revise it to align with the current abstract. And I would also suggest that include "inflammation" on the title as I think this work is more than just studying "aging".*

Response. We agree that our results support a necroptosis-independent role of the RIPK3-MLKL axis rather than the effect of its sublethal activation. While we also agree that inflammation is another keyword to consider in the title, we think inflammation is just one of the stressors that induce age-related HSC changes via non-necroptotic MLKL activation in HSCs. Now that we have established the causality between MLKL and mitochondrial impairment by the above experiments, we feel that including this notion helps the title better represent the scope of the revised manuscript. Thus, we have changed the title to “non-necroptotic MLKL function damages mitochondria and promotes hematopoietic stem cell aging” and revised the text accordingly.

Reviewer #3

We thank the reviewer for the positive and insightful comments. We appreciate the recognition of our work’s significance in revealing a non-canonical role of MLKL in HSC regulation. We have carefully

considered all points to further strengthen the manuscript, with particular emphasis on additional analyses of NEO-1, including transplantation experiments.

Comment 1. *The manuscript by Yamada et al. describes the role of Mlkl in hematopoietic stem cells (HSC). Authors showed that Mlkl is activated in HSC and multipotent progenitor (MPP) pool upon inflammation using FRET based sensor that monitor MLKL translocation to membrane. However, they showed that this activation does not lead to cell death of HSCs, as expected by the known function of Mlkl in necroptosis. Next, authors performed series of experiments on Mlkl^{-/-} mice, and demonstrated that such mice do not show many established hallmarks of HSC aging or strong inflammation-driven exhaustion of HSC function. Finally, the authors showed the link of Mlkl with age-related changes in HSCs and demonstrated that Mlkl^{-/-} HSCs has more ATP production and sustained glycolytic flux. Significance and innovation: In the field of HSC aging there are many phenotypes reported that show premature aging of HSC. However, there very few studies demonstrating phenotypes that protect HSCs from acquiring aging hallmarks, like myeloid-biased or stem cell pool expansion with reduced reconstitution potential. Therefore, in my opinion the presented work is significant contribution to the field. While I am not an expert in the field of programmed cell death and necroptosis, the observation of the Mlkl translocation to membrane without induction of necroptosis seem to be not well studied, with some prior papers (eg. Vucur et al Immunity 2023) cited. My major question/remarks: First is the conclusion about “sublethal necroptosis” stated within the title and text. While the term “sublethal necroptosis” has been used in field previously, in my opinion it might be misleading – either there is necroptosis, and the cell is dying or there is just necroptosis-independent role of Mlkl. The fact that the used FRET sensor shows Mlkl translocation to the membranes may not necessarily means it is involved in membrane rupture. Now, given the novel link with mitochondrial energy production this together may indicate an unrecognized function of Mlkl, that is not related to necroptosis. eg. membrane trafficking. Therefore, I am not fully convinced with the term sublethal necroptosis.*

Response. We appreciate the reviewer’s thoughtful comments. We agree that our results support a necroptosis-independent role of the RIPK3-MLKL axis rather than the effect of its sublethal activation. To address this and other points raised by the reviewer, we have now revised the title to “non-necroptotic RIPK3-MLKL damage mitochondria and promotes hematopoietic stem cell aging by damaging mitochondria.”

Comment 2. *Second, I think it is not logically correct to state that “Sublethal necroptosis underlies hematopoietic stem cell aging” based on the observation of Mlkl deficiency result is resistance to age-related changes in HSC. I think, based on what is presented I would state that loss of MLKL protects against hallmarks of HSC aging, what is still an important observation.*

Response. We apologize for any unintended consequences arising from the manuscript’s title and wording, which may have been misleading. We totally agree that the partial restoration of aging hallmarks in *Mlkl^{-/-}* HSCs rather indicates the RIPK3-MLKL axis is one of the mechanisms that mediate HSC aging, and this was indeed the conclusion we highlighted in the Discussion section of the original manuscript. As detailed below in our response to the reviewer’s comment 3, we have performed additional experiments using isolated mitochondria and recombinant MLKL protein, and demonstrated that MLKL directly impairs mitochondrial function (Fig. 6h). To better reflect these notions, we have now changed the title to “non-necroptotic MLKL function damages mitochondria and promotes hematopoietic stem cell aging” and revised the text accordingly.

Comment 3. *The article open next important questions. How the Mlkl affects mitochondria, what is role here? However I fully understand that this a wide research question and may be beyond the current study and was also stated by the authors in the discussion. But one thing I would suggest for consideration is to add that at this point the observed reversal of mitochondrial function may be the reason or the effect of the aging protection.*

Response. We thank the reviewer for this insightful suggestion that gave us an opportunity to gain mechanistic insights as to how MLKL impairs mitochondrial function. We first confirmed that N-terminal FLAG-tagged MLKL (FLAG-MLKL), which lacks the capacity to form pores on the cell membrane, can still target mitochondria in HSCs using proximity ligation assay PLA (Supplementary Fig. 8b). To determine whether pore-forming function of MLKL directly impairs mitochondrial membrane potential, mitochondria were isolated from mouse liver and incubated *in vitro* with either (i) N-terminal four-helix bundle domain of recombinant mouse MLKL protein (MLKL-NTD) that oligomerizes and forms pores on the cell membrane without stimulation, (ii) N-terminal FLAG-tagged MLKL-NTD (FLAG-MLKL-NTD) that retains oligomerization capacity but cannot perform pore formation on the cell membrane, or (iii) C-terminal pseudokinase domain of recombinant mouse MLKL protein (MLKL-CTD) that does not oligomerize or form pores on the cell membrane (Hildebrand JM et al. PNAS 2014, PMID: 25288762). Mitochondria were then stained with Tetramethylrhodamine methyl ester (TMRM) as well as MitoTracker green, and TMRM signals within MitoTracker-positive population were analyzed by flow cytometry. TMRM signals in mitochondria were significantly reduced by incubation with MLKL-NTD, and to a lesser extent by FLAG-MLKL-NTD, but not by MLKL-CTD (Fig. 6h). These results indicate that activated MLKL directly impairs mitochondrial membrane potential via mechanisms both dependent and independent of its pore-forming capacity.

Comment 4. *Finally, I find very interesting that the protection of aging in Mlkl^{-/-} mice is not linked to reversal of expansion of the expansion of myeloid-biased HSC defined as CD41⁺. Now, given the authors transplanted the whole pool of HSCs that means that either Mlkl-deficiency protects the “balanced” fraction of HSCs during aging or reverse the myeloid and aging phenotype of “myeloid-biased” HSCs. Given it is possible now to prospectively transplant myeloid-biased HSCs (work by Ross et al. using Neo-1 as marker, Nature 2024, Weissman lab) would it be possible to answer this question?*

Response. We thank the reviewer for this important comment. Additional experiments with PBS- or polyinosinic-polycytidylic acid (pIC)-treated *Mlkl*^{-/-} mice revealed that MLKL deficiency significantly attenuates pIC-induced NEO-1 upregulation on HSCs (Fig. 2d). To further assess whether activation of the RIPK3-MLKL axis affects age-related HSC subpopulations, we utilized MLKL-SA2 knock-in mice (JAX #039024) where MLKL cannot be activated by RIPK3-dependnet phosphorylation due to the two serine-to-alanine substitutions at codons 345 and 347 of the *Mlkl* locus (Yeap HW et al. EMBO J 2025, PMID: 40128366), and treated them with PBS or pIC. In addition to NEO-1, we also assessed surface expression of P-selectin and GPR183, which were reported to enrich myeloid-biased (Flohr Svendsen A et al. Blood 2021, PMID: 33876187) and less differentiating (Totani H et al. Nat Aging 2025; PMID: 40050412) subpopulations in aged HSCs, respectively. The results revealed that inactivation of the RIPK3-MLKL axis significantly attenuated pIC-induced upregulation of NEO-1, and to a lesser extent GPR183, but not P-selectin, on HSCs (Fig. 2e, Supplementary Fig. 2e-f). As the total number of HSCs did not differ before and after acute pIC injection (Fig. 2c), this indicates that the RIPK3-MLKL axis promotes transition from NEO-1⁻ to NEO-1⁺ states. Transplantation of NEO-1⁻ and NEO-1⁺ HSCs isolated from pIC-treated WT and *Mlkl*^{SA2} mice further revealed a trend toward improved engraftment of *Mlkl*^{SA2} NEO-1⁻ HSCs, but not NEO-1⁺ HSCs, compared to WT counterparts (Supplementary Fig. 3e-i), indicating

selective protection of NEO-1⁻ HSCs by blockade of the RIPK3-MLKL axis. Thus, we concluded that the RIPK3-MLKL axis primarily promotes age-related functional decline in NEO-1⁻ lineage-balanced HSCs.

Comment 5. *Research strategy: In my opinion all HSC assays are well and robustly performed, well-presented and I have no doubts about the clear and significant effect of the Mlkl-deficiency on HSC aging protection. There are few places where I would suggest some more precise description (below).*

Response. We thank the reviewer for the positive assessment of our research strategy and HSC assays. We have revised the manuscript according to the following comments.

Other comments:

Comment 6. *Extended Fig 1 C – the quality of the blot is poor, not sure if it is needed to include it – maybe the IHC is more informative*

Response. In accordance with the reviewer’s suggestion, we have removed the Western blot result and deleted the corresponding text.

Comment 7. *Authors wrote observed lower engraftment potential and a tendency toward reduced lymphopoietic potential of FRET/CFPhi HSCs compared to FRET/CFPlo HSCs. (Fig 1d-f) But this mostly B lineage at the cost of myelo, T cells are not changed here.*

Response. We have revised the text accordingly (line 86).

Comment 8. *In pIC experiments authors stated that Mlkl deficiency significantly sustained the lymphoid potential, but to be precise is still only B lineage, and T lin even decreased.*

Response. We have revised the text accordingly (lines 115–116).

Comment 9. *Fig. 1 – it is stated that Mlkl activation is preferential in HSC, but the data shows it is also observed in MPPs to the same extend.*

Response. We have revised the text accordingly (lines 75–76).

Comment 10. *The same is in the Fig. 3a – statement that the activation is preferential to HSCs would require comparison to more populations (eg. MPPs) not only to MP*

Response. Additional analysis revealed that FRET signals were significantly elevated in MPP2 and MPP3/4 but not in CLP upon 5-FU treatment (Fig. 3b, Supplementary Fig. 4a). We have revised the text to reflect the results (lines 137–139).

Comment 11. *Authors discussed the changes of the mitochondria with aging as “HSC memory”. While I found the concept of HSC memory very valid and interesting, in my opinion the term memory makes sense when the memory govern some advantage to secodanry stimulai/challenge. Otherwise for me it is just another aging hallmark/feature. I can’t see what kind of “memory advantage” would the observed decline in mitochondria decline would provide.*

Response. We agree that the term “memory” could imply an adaptation mechanism that is primarily designed for cells to improve their outcome in response to secondary stimuli. While accumulating damaged mitochondria could be an adaptive HSC response to prevent their unlimited self-renewal as discussed elsewhere (Hinge A et al. Cell Stem Cell 2020, PMID: 32059807), this concept has not formally been proven yet. We have thus rephrased the text in Supplementary Fig. 8h and the Discussion section (lines 245–253) to infer cumulative cellular damage that limits HSC function after stress.

Point-by-point responses to reviewers' comments:

We thank the editor and reviewers for the in-principle acceptance of our manuscript. As no further revisions were requested at this stage, we confirm that no additional changes have been made beyond those already incorporated during the previous revision cycle, except for minor data corrections and brief textual clarifications. We are grateful for the constructive feedback that strengthened the study.

Reviewer #1

***Comment.** In the revised study by Yamada et al., the authors have comprehensively addressed the concerns raised during the previous review round by incorporating additional experiments and new analyses. These additions substantially strengthen the manuscript and enhance the clarity of its conclusions. Notably, the inclusion of experiments investigating MLKL in mitochondrial function and the role of RIPK3–MLKL in age-related HSC functional decline, through the analysis of NEO-positive and NEO-negative HSCs, represents a significant improvement. Overall, the revisions have increased the quality and impact of the study. I am satisfied with the authors' responses and the new data presented.*

Response. We sincerely thank the reviewer for the positive assessment and for recognizing the added experiments and analyses, including the MLKL mitochondrial function assays and RIPK3–MLKL evaluations in Neo-1⁺ and Neo-1⁻ HSCs.

Reviewer #2

***Comment.** Accept*

Response. We thank the reviewer for her/his support of the manuscript.

Reviewer #3

***Comment.** The authors have addressed all of my comments. I find the observation of reversed Neo-1 expression and the analysis of myeloid-biased HSCs following pIC treatment particularly interesting and representing an added value to the manuscript. I also acknowledge that the authors performed Neo-1⁺/Neo-1⁻ HSC transplantation experiments, which support the conclusion that RIPK3–MLKL blockade acts on the Neo-1 fraction; however, the number of mice included in these experiments is relatively limited. Nevertheless, in my opinion, the revision is comprehensive and has significantly improved the manuscript.*

We thank the reviewer for the constructive feedback and positive evaluation. We acknowledge the comment regarding the limited number of mice in the transplantation experiments and have added a brief clarification in the Results section (line 122).